# Characterization of elusive rhamnosyl dioxanium ions and their application in complex oligosaccharide synthesis

Peter H. Moons [1,3], Floor ter Braak[1,3], Frank F. J. de Kleijne[1,3], Bart Bijleveld [1], Sybren J. R. Corver[1], Kas J. Houthuijs [2], Hero R. Almizori[1], Giel Berden [2], Jonathan Martens [2], Jos Oomens[2], Paul B. White[1] ✉ & Thomas J. Boltje [1] ✉

Attaining complete anomeric control is still one of the biggest challenges in carbohydrate chemistry. Glycosyl cations such as oxocarbenium and dioxanium ions are key intermediates of glycosylation reactions. Characterizing these highly-reactive intermediates and understanding their glycosylation mechanisms are essential to the stereoselective synthesis of complex carbohydrates. Although C-2 acyl neighbouring-group participation has been well-studied, the reactive intermediates in more remote participation remain elusive and are challenging to study. Herein, we report a workflow that is utilized to characterize rhamnosyl 1,3-bridged dioxanium ions derived from C-3 *p*-anisoyl esterified donors. First, we use a combination of quantum-chemical calculations and infrared ion spectroscopy to determine the structure of the cationic glycosylation intermediate in the gas-phase. In addition, we establish the structure and exchange kinetics of highly-reactive, low-abundance species in the solution-phase using chemical exchange saturation transfer, exchange spectroscopy, correlation spectroscopy, heteronuclear single-quantum correlation, and heteronuclear multiple-bond correlation nuclear magnetic resonance spectroscopy. Finally, we apply C-3 acyl neighbouring-group participation to the synthesis of complex bacterial oligosaccharides. This combined approach of finding answers to fundamental physical-chemical questions and their application in organic synthesis provides a robust basis for elucidating highly-reactive intermediates in glycosylation reactions.

Complete stereocontrol over glycosidic bond formation remains one of the main challenges in the chemical synthesis of carbohydrates. In a chemical glycosylation reaction, a glycosyl donor is activated by an electrophilic promotor and reacts with a nucleophilic hydroxyl group on a glycosyl acceptor to afford a glycosidic bond. The nucleophile can add from the α- or β-face of the glycosyl donor, thereby forming α- or β-diastereoisomers, respectively. One of the most common and

reliable ways to control the stereochemical outcome of glycosylation reactions is the utilization of neighbouring group participation (NGP) of C-2 acyl groups on glycosyl donors[1-3]. Upon activation of such glycosyl donors, the C-2 acyl group can engage in NGP affording a *cis*-fused bicyclic dioxolanium ion intermediate that reacts in a stereo-specific manner with glycosyl acceptors to afford 1,2-*trans* products. The participation of acyl groups positioned on the C-3, C-4 or C-6

[1]Synthetic Organic Chemistry, Institute for Molecules and Materials, Radboud University Nijmegen, Heyendaalseweg 135, 6525 AJ Nijmegen, The Netherlands. [2]FELIX laboratory, Institute for Molecules and Materials, Radboud University Nijmegen, Toernooiveld 7, 6525 ED Nijmegen, The Netherlands. [3]These authors contributed equally: Peter H. Moons, Floor ter Braak, Frank F. J. de Kleijne. ✉e-mail: paul.white@ru.nl; t.boltje@ru.nl

hydroxyl groups of a glycosyl donor has also been suggested to direct the stereoselectivity of glycosylation reactions[4–10]. However, whether the observed effects on stereoselectivity can be attributed to NGP of the acyl group, or by the introduction of other stereoelectronic effects, is a subject of much debate[11]. Whereas indirect evidence of this participation has been reported[4,12–15], direct evidence of the bridged dioxanium or dioxepanium reactive intermediates under standard glycosylation conditions is scarce. The highly reactive and unstable nature of such reactive intermediates severely complicates their characterization.

Recently, we and others reported the spectroscopic evidence of bridged intermediates[16–21], including those resulting from the NGP of C-3 and C-4 esters on glucosides, galactosides and mannosides, using gas-phase infrared ion spectroscopy (IRIS)[22]. Density functional theory (DFT) calculations were used to assign the IRIS spectra of glycosyl cations and provide insight into the stability and conformational energy landscape (CEL) of these species. These experiments suggested that a C-3 acyl group on mannosyl donors can readily access the conformation needed for participation, leading to a stable dioxanium ion. Subsequent solution-phase glycosylations of mannosides carrying a single acyl group at C-3 proceeded with very high α-selectivity compared to derivatives containing a non-participating group at C-3. This observation is consistent with NGP of the C-3 ester, which shields the β-face and only allows nucleophilic addition to the α-face. More recently, we characterized the mannosyl dioxanium ion intermediate in solution under relevant reaction conditions using variable temperature NMR (VT-NMR) and established its exchange kinetics with the α-triflate intermediate using chemical exchange saturation transfer (CEST) NMR[23]. The observed rapid exchange kinetics likely fall within the boundaries expected for a Curtin-Hammett scenario, which plausibly explains the α-selective nature of these mannosyl donors[24]. In sharp contrast, the corresponding glucosyl donor containing the same C-3 acyl group did not show dioxanium ion formation and was unselective in glycosylation reactions. The interesting observation that C-3 acyl groups on manno-type donors, which contain an axial C-2 substituent, are able to engage in anomeric stabilization triggered us to investigate other manno-type sugars. Nicolaou et al. showed that the activation of N-acetyl D-mycosamine donors (3-NHAc D-rhamnose) led to the formation of the corresponding bridged oxazoline[25]. More recently, Lei et al. demonstrated that the installation of a C-3 acyl group on rhamnosyl donors led to very high α-selectivity, but conclusive evidence of the reactive intermediates involved in C-3 acyl participation was not provided[6]. Whether C-3 acylation results in α-selective glycosylation via NGP or inductive effects remains a topic of debate as no direct evidence for the 1,3-bridged rhamnosyl dioxanium ion has been provided to date.

Herein, we report a workflow for the characterization of these elusive intermediates. Rhamnosyl dioxanium ions, resulting from C-3 acyl participation, were characterized in the gas-phase using IRIS and in the solution-phase under relevant glycosylation conditions using CEST NMR. In order to investigate the stereo-directing ability of C-3 acyl participation in the context of the total synthesis of a complex oligosaccharide, we prepared the O-antigen repeating unit of *Burkholderia pseudomonas (B. pseudomonas)* and *Serratia marcescens (S. marcescens)*. This work provides a workflow that was utilized to confirm that C-3 acyl NGP in rhamnosides proceeds through anomeric stabilization, forming a dioxanium ion intermediate. Finally, we demonstrate that NGP of C-3 acyl groups can be employed as reliable method in the total synthesis of complex bacterial oligosaccharides to construct glycosidic linkages with complete α-selectivity.

## Results

To validate the stereodirecting capability of C-3 acyl groups on rhamnosyl donors, we first investigated the glycosylation behavior of rhamnosyl donors **1** and **2** under pre-activation and pre-mix conditions (Fig. 1; Supplementary Table 1). $^1J_{C,H}$-couplings were measured with $^{13}C$-coupled heteronuclear single quantum coherence (HSQC) spectroscopy in order to assign the anomeric stereochemistry (Supplementary Fig. 3)[26,27]. Next, α:β selectivity was determined using quantitative HSQC spectroscopy (Supplementary Fig. 3)[28,29]. The donor carrying a C-3 anisoyl group (**1**) was very α-selective whilst the benzylated control (**2**) displayed poor selectivity under both activation conditions. While selectivity may be affected by promoter choice[30], the benzylated rhamnosyl donor likely reacts via glycosyl triflate intermediates of the solvent-separated ion pair (SSIP), thus resulting in poor selectivity independent of the promoter system used. The introduction of the C-3 anisoyl group gives rise to the additional possibility of forming a bridged dioxanium ion that would afford the α-rhamnoside. An anomeric mixture was indeed observed for rhamnosyl donor **2**. In contrast, glycosylations with rhamnosyl donor **1** were highly α-selective irrespective of activation conditions, which prompted us to further study the glycosylation mechanism by identifying the reaction intermediates.

To investigate the existence of C-3 acyl participation in rhamnosyl donors with IRIS and CEST NMR, the C-3 benzoylated thioglycoside donor **3** was prepared (Fig. 2). The other hydroxyl groups were protected as benzyl ethers. Furthermore, perbenzylated rhamnosyl donor **2** was prepared as a control compound. First, L-rhamnose was peracetylated by treatment with acetic anhydride (Ac$_2$O) in CH$_2$Cl$_2$/pyridine (9:1, v/v). A subsequent reaction with thiophenol and BF$_3$·OEt$_2$ afforded the corresponding thioglycoside, which was deacetylated using K$_2$CO$_3$ in a mixture of tetrahydrofuran (THF) and methanol (1:1, v/v) to give thioglycoside **9**. Selective protection of the C-3 position using stannylene acetal-mediated chemistry afforded 2-methylnaphthyl derivative **10**, which was benzylated under standard conditions to yield rhamnoside **11**. Subsequent oxidative removal of the 2-methylnaphthyl ether using 2,3-dichloro-5,6-dicyano-p-benzoquinone (DDQ) provided the corresponding C-3 hydroxyl precursor **12** in a moderate yield over three steps. At this stage, the C-3 participating group was installed. We selected the benzoyl group as it is structurally very similar to the benzyl ether control, while it differs significantly in its electronic properties and its ability to stabilize an oxocarbenium ion[31,32]. Notably, a CEST profile is obtained by selecting a reporter peak (the $^{13}C$-labeled α-triflate) and then plotting it against the degree of saturation transfer. Because the latter is affected by variables like the observed nucleus, the frequency difference between exchanging species and the exchange rate between exchanging species, a CEST profile may be different depending on factors that influence the exchange rate, such as the sterics, electronics and protecting group pattern[33]. We have seen that changes in protecting groups (methyl, benzyl, benzylidene) can have profound effects on the exchange kinetics in our previous work on mannosyl dioxanium ions[23,33]. Accordingly, switching from a mannoside to a 6-deoxymannoside, namely rhamnose, will likely affect the exchange kinetics. In addition, we envisioned that varying the electron density within an acyl group can be used as a tool to change the exchange kinetics. The latter may be necessary because the CEST NMR viewing window requires the difference in resonance frequency between two exchanging species to be greater than their chemical exchange rate ($\Delta\omega > k_1 + k_{-1}$)[33]. Therefore, we synthesized the p-anisoyl derivative **1** in order to further modulate the stability, population, and exchange kinetics of the expected dioxanium ion intermediate. We expected the enhanced electron-donating character of the p-methoxy group to stabilize the cationic species, along with further stabilizing the dioxanium ion through resonance. For the NMR experiments, $^{13}C$-labeled esters **4** and **5** were prepared to increase the sensitivity of the $^{13}C$ CEST experiments. The C-3 acyl-protected rhamnosyl donors **1**, **3** and their $^{13}C$-labeled derivatives **4** and **5** were prepared using a Staab or Steglich esterification.

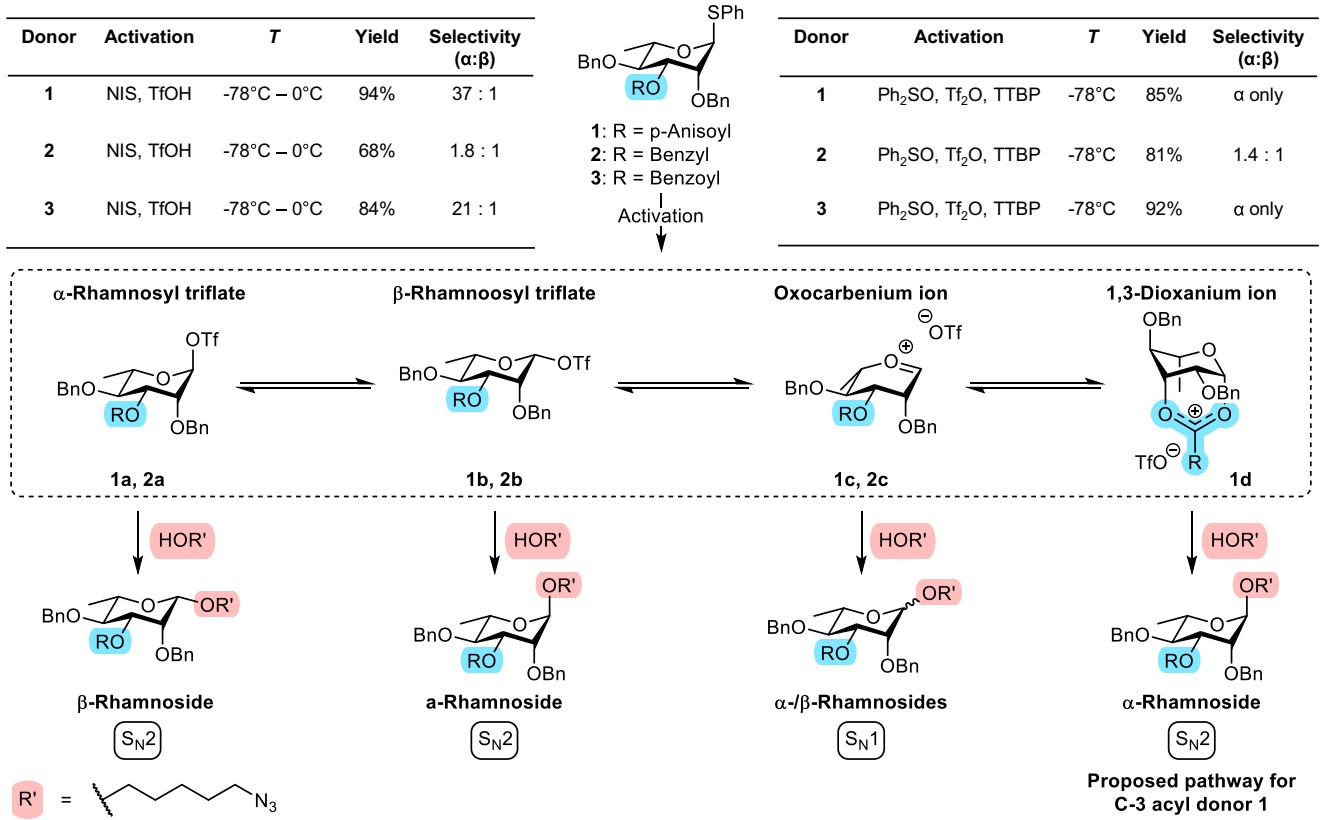

| Donor | Activation | *T* | Yield | Selectivity (α:β) |
|---|---|---|---|---|
| 1 | NIS, TfOH | -78°C – 0°C | 94% | 37 : 1 |
| 2 | NIS, TfOH | -78°C – 0°C | 68% | 1.8 : 1 |
| 3 | NIS, TfOH | -78°C – 0°C | 84% | 21 : 1 |

**1**: R = p-Anisoyl
**2**: R = Benzyl
**3**: R = Benzoyl

| Donor | Activation | *T* | Yield | Selectivity (α:β) |
|---|---|---|---|---|
| 1 | Ph₂SO, Tf₂O, TTBP | -78°C | 85% | α only |
| 2 | Ph₂SO, Tf₂O, TTBP | -78°C | 81% | 1.4 : 1 |
| 3 | Ph₂SO, Tf₂O, TTBP | -78°C | 92% | α only |

**Fig. 1 | Rhamnosylation mechanism.** Proposed glycosylation intermediates **1a–d** and **2a–c** for rhamnosyl donors **1** and **2**, respectively. 5-Azidopentanol was used as glycosyl acceptor. α/β-selectivity was determined using quantitative HSQC prior to purification with silica-flash column chromatography[28,29].

## Characterization of the 1,3-dioxanium ion in the gas-phase

To explore the possibility of dioxanium ion formation on rhamnosyl donors **1** and **3**, we investigated the gas phase structure of their corresponding glycosyl cations. First, we generated rhamnosyl cations from precursors **1** and **3** using a previously reported tandem mass spectrometry (MS/MS) scheme in the gas-phase (Supplementary Fig. 1-2)[16,34]. The structure of the gas-phase ions was probed using IRIS with the FELIX infrared free-electron laser operating between 750 and 1850 cm⁻¹. We have previously demonstrated that this frequency range contains highly diagnostic vibrational bands that distinguish between oxocarbenium and dioxanium ions[22,35]. The oxocarbenium ion is characterized by a $C_1=O_5^+$ carbonylonium stretch (~1600 cm⁻¹) with preservation of the C = O carbonyl stretch (~1750 cm⁻¹)[36]. In contrast, absence of the carbonylonium C = O⁺ stretch indicates the formation of the bicyclic glycosyl dioxanium ion. The formation of this bicycle is diagnosed by the presence of O-C⁺-O and C⁺-C_Ar stretching modes at ~1520 and ~1420 cm⁻¹, respectively. Structural assignment of observed spectra (black) is further supported by comparing them with DFT-calculated IR spectra (MP2/6-31 + + G(d,p)//B3LYP/6-31 + + G(d,p)) using a previously described workflow[22].

Well-resolved IR spectra of the isolated ions derived from **1** and **3** were obtained (Fig. 3, black line). Comparison with DFT calculations revealed the presence of dioxanium structures **1⁺B** and **3⁺B**, which result from C-3 ester participation. All major experimental peaks were assigned based on their excellent agreement with the theoretical IR spectra. Most notably, the absence of the C = O carbonylonium[36] stretch excludes the presence of oxocarbenium ion isomers **1⁺A** and **3⁺A**. Moreover, a better match is presented for the major dioxanium ion stretches (Fig. 3B,D; C⁺-C_Ar, O-C⁺-O, and C-H_OMe) rather than for the major oxocarbenium stretches (Fig. 3A,C) for both the benzoyl and the *p*-anisoyl donors.

## Characterization of the 1,3-dioxanium ion in the solution-phase

The IRIS experiments confirm that the formation of dioxanium ions is geometrically feasible in rhamnose donors **1** and **3** in the gas-phase and in the absence of a counterion. The detection of these intermediates under relevant glycosylation conditions is much more challenging as the presence of triflate counter ions likely causes a rapid equilibrium shifting from the dioxanium ion to the more stable α-rhamnosyl triflate. Therefore, we investigated the existence of the bridged species in the solution-phase under relevant glycosylation conditions using ¹³C CEST NMR[23,37,38]. To this end, substrates **4** and **5** containing a ¹³C label on the C-3 benzoyl carbonyl carbon were employed. These substrates benefit from a large difference in frequency (Δω) between exchanging species compared to ¹³C labelling at C-1. In addition, ¹³C carbonyl carbons assure a slower relaxation rate (R₁) as they have no protons to assist in dipolar relaxation, thus leading to a more ideal relaxation time relative to the exchange rate[39].

Rhamnosyl donors **4** and **5** were activated at -80 °C using diphenyl sulfoxide (Ph₂SO) and triflic anhydride (Tf₂O) in the presence of non-nucleophilic base, 2,4,6-tri-*tert*-butylpyrimidine (TTBP)[40]. Both rhamnosyl donors were activated at -80 °C, forming the α-triflate after being heated to -40 °C (Supplementary Fig. 6-7). After cooling the rhamnosyl α-triflates down to -80 °C, no formation of a dioxanium ion was observed using conventional 1D ¹H and ¹³C NMR or 2D HSQC and HMBC NMR (Supplementary Fig. 8-11). ¹³C CEST NMR was then employed to scan for low-abundance transient reaction intermediates at -80 °C. Based on the ¹³C CEST profile, no dioxanium ion formation was observed for C-3 benzoyl-protected rhamnosyl donor **5** (Fig. 4A). Interestingly, ¹³C CEST NMR revealed that the C-3 anisoyl rhamnosyl α-triflate was in chemical exchange with a rapidly exchanging species bearing a more downfield signal ($\delta_C$ = 174.6 ppm) compared to the carbonyl carbon ($\delta_C$ = 165.2 ppm) (Fig. 4A). The detected chemical shift

suggests the formation of the elusive dioxanium ion (Supplementary Fig. 12), as observed previously for mannosyl 1,3-dioxanium ions[23,33]. This was only observed for the anisoyl donor and supports the hypothesis that stabilization of reactive dioxanium intermediates

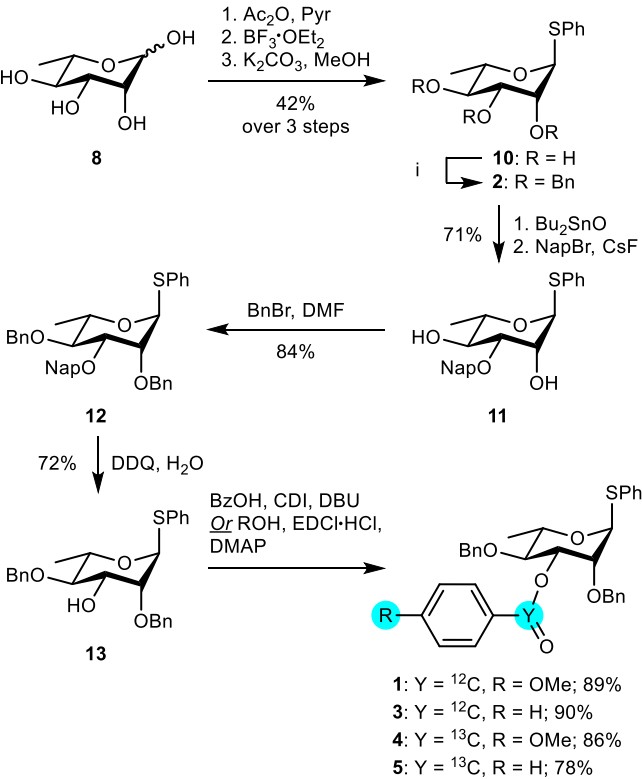

**Fig. 2 | Rhamnosyl donor 1 – 5 synthesis.** The C-3 hydroxyl is selectively protected with a 2-methylnapthyl ether prior to deprotection and functionalization with a $^{12}$C- or $^{13}$C-labeled acyl group.

through resonance leads to an increase in dioxanium ion population and a decrease in exchange rate, thereby falling within the window required for CEST NMR ($k_{ex} < \Delta\omega$). Adding glycosyl acceptor **19**, *p*-toly 2,3,6-tri-*O*-benzyl-1-thio-β-D-glucopyranoside, resulted in full consumption of the α-triflate within five minutes. The disaccharide product was confirmed to be the α-rhamnoside, based on its $^1J_{CH}$ coupling (Supplementary Fig. 13-15)[41].

In our previous work on C-3 acyl mannosyl donors, we observed that 1,3-dioxanium ions could be generated at -40 °C in the absence of TTBP to afford a population high enough for characterization using 2D-NMR[23]. Therefore, we activated rhamnosyl donors **4** and **5** at -40 °C in the absence of TTBP. After 3 hours, the solution was cooled down to -80 °C. This time, $^{13}$C CEST NMR spectroscopy did provide evidence for C-3 benzoyl participation (Supplementary Fig. 16). However, the population was too low to detect it with $^1$H, $^{13}$C, and HSQC NMR spectroscopy. In contrast, the 1,3-dioxanium ion derived from C-3 anisoyl participation was clearly visible in $^{13}$C CEST, $^1$H and $^{13}$C spectra (Supplementary Fig. 17-18). Moreover, $^1$H-$^{13}$C HMBC NMR confirmed the 1,3-bridge through cross-peaks from the anomeric proton ($\delta_H = 6.38$ ppm) and H-3 ($\delta_H = 5.53$ ppm) to the $^{13}$C-labelled carbonyl ($\delta_C = 174.6$ ppm) (Fig. 4B). In addition, $^1$H-$^{13}$C HSQC NMR reveals that the anomeric proton at $\delta_H = 6.38$ ppm is coupled to a resonance at $\delta_C = 103.4$ ppm (Supplementary Fig. 17), which is in good agreement with the anomeric carbon signal observed in mannosyl dioxanium ions.

Furthermore, $^1$H-$^1$H COSY NMR suggests that the dioxanium ion adopts a $^1C_4$-chair conformation because a characteristic W-coupling pattern between H-1 and H-3 was observed (Fig. 4B). Further characterization of the dioxanium ion was limited by small vicinal axial-equatorial or equatorial-equatorial coupling and by peak broadening associated with temperature and exchange. Conclusively, the combined 1D and 2D NMR experiments are in accordance with the dioxanium ion structure identified by IRIS.

Having established that the α-triflate is in chemical exchange with the dioxanium ion, we set out to further study the exchange mechanism of the rhamnosyl α-triflate **4a**. We recently demonstrated that $^{19}$F exchange spectroscopy (EXSY) NMR can be used to distinguish between the intramolecular and intermolecular glycosyl triflate intermediate

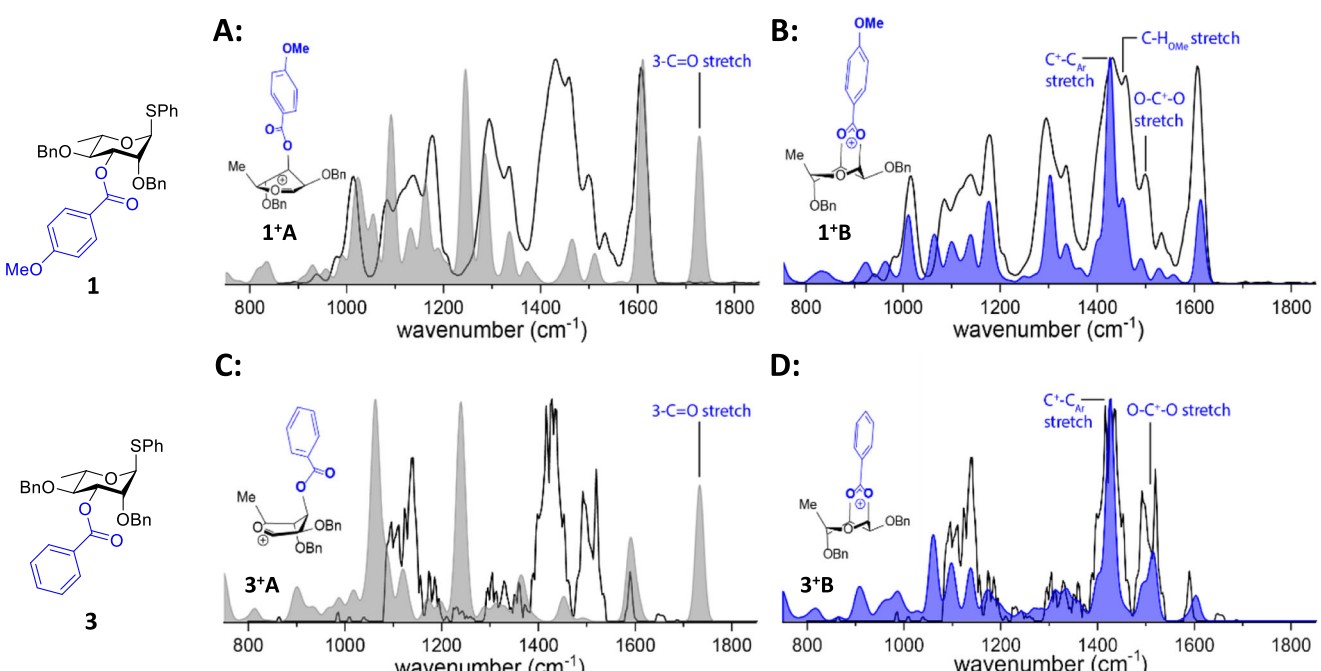

**Fig. 3 | IR ion spectra of rhamnosyl cation 1$^+$ and 3$^+$.** Comparison of the measured IR-ion spectrum (black line) with the calculated spectra (filled) of: (**A**, **C**) oxocarbenium ions 1$^+$A and 3$^+$A (grey); (**B**, **D**) 1,3-bridged dioxanium ions 1$^+$B and 3$^+$B (blue).

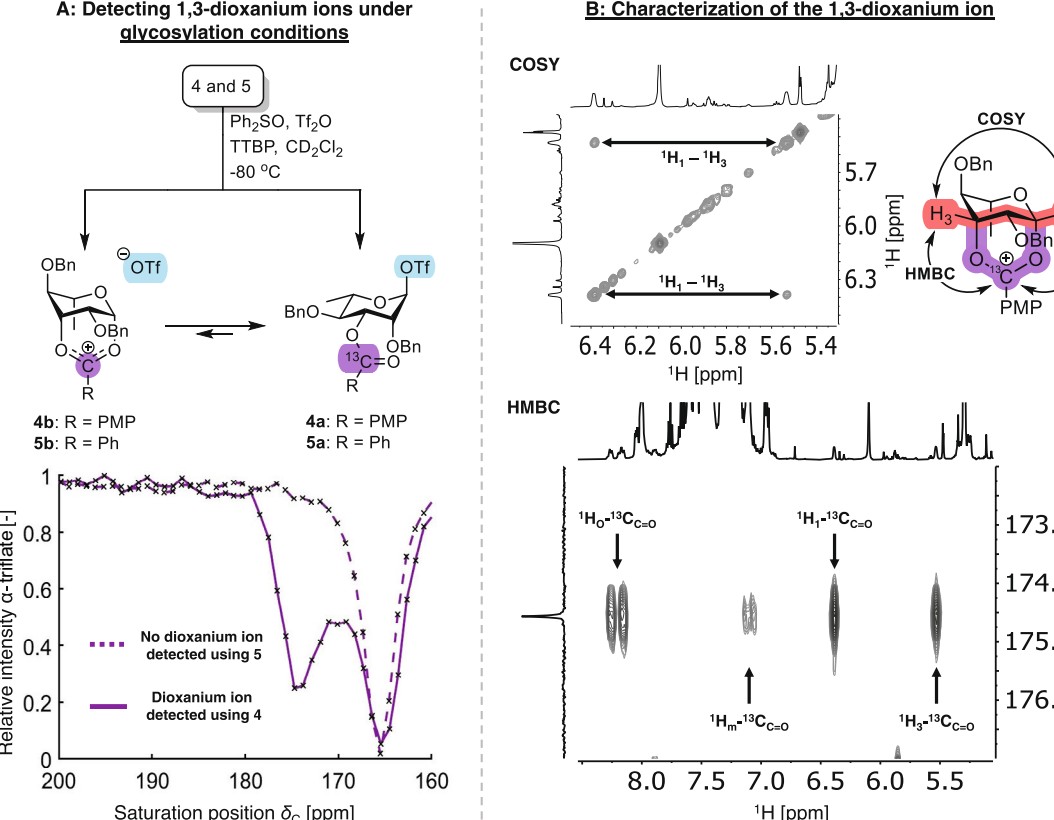

**Fig. 4 | VT-NMR experiments following activation of rhamnosyl donors 4 and 5.** VT-NMR experiments (-80 °C) to study: **A**) Presence of 1,3-dioxanium ion under representative glycosylation conditions; **B**) Structure of the dioxanium ion under glycosylation conditions without TTBP. $^1H_1$-$^{13}C_{C=O}$ is the cross-peak of H-1 to the $^{13}C$ labelled carbon; $^1H_3$-$^{13}C_{C=O}$ is the cross-peak of H-3 to the $^{13}C$-labeled carbon; $^1H_o$-$^{13}C_{C=O}$ and $^1H_m$-$^{13}C_{C=O}$ are the cross peaks of the ortho- and meta-protons of the anisoyl group to the $^{13}C$-labeled carbon, respectively. PMP = *para*-methoxyphenyl.

exchange mechanisms (Fig. 5A)[33]. After activation, both the α-rhamnosyl triflate and unbound triflate anion are observed as strong signals with $^{19}F$ NMR. Selective excitation of either resonance can be used to determine what the selected resonance is in chemical exchange with. Moreover, exchange rates can be derived using the initial rate approximation (Supplementary Fig. 4). The exchange mechanism can be determined by taking into account that the rhamnosyl α-triflate can dissociate into free triflate (·OTf) in three possible mechanisms, namely: 1) dissociation of the α-triflate, affording the rhamnosyl SSIP; 2) direct participation of the carbonyl without forming a SSIP; 3) $S_N2$-like displacement by free triflate in solution, affording the β-rhamnosyl triflate. As only the latter is influenced by free triflate concentration, studying the exchange rate at different triflate concentrations provides insight into the influence of intramolecular or intermolecular stabilization of the glycosyl cation.

After activation of rhamnosyl donor **4** under standard glycosylation conditions, the exchange rate of α-triflate dissociation ($R_{α → OTf, EXSY}$) was measured using $^{19}F$ EXSY (Supplementary Fig. 23). Next, the rate of dioxanium ion formation from the α-triflate ($R_{α → d, CEST}$) was determined using $^{13}C$ CEST NMR (Supplementary Fig. 27)[42]. Finally, the influence of triflate concentration was evaluated by adding increasing amounts of tetrabutylammonium triflate (TBAT) (Supplementary Fig. 23-30). These results clearly demonstrate that the α-triflate exchanges in a mixed mechanism resulting from stabilization by both intramolecular dioxanium ion formation and intermolecular β-rhamnosyl triflate formation. However, the influence of intramolecular stabilization is the dominating factor because $R_{α → OTf, EXSY} ≈ R_{α → d, CEST}$ (1.3 +/- 0.03 *vs*. 1.2 +/- 0.04) under standard glycosylation conditions that include no additional free triflate (Fig. 5B). The influence of a β-triflate reactive species appears non-existent under these conditions and hence we propose that the

α-selective glycosylation proceeds through the dioxanium ion rather than the β-triflate.

## Application in complex oligosaccharide synthesis

Having confirmed rhamnose C-3 acyl NGP in glycosylation reactions, we investigated the robustness of this effect in complex oligosaccharide synthesis. We identified the *O*-antigen of the opportunistic human pathogens *Burkholderia pseudomonas* and *Serratia marcescens* as a suitable target. They share a structure of the α-(1,3)-rhamnopyranosyl-α-(1,4)-glucopyranosyl repeating unit in the *O*-antigen of several serogroups[43,44]. *B. pseudomonas* is involved in cystic fibrosis and onion rot, whereas *S. marcescens* is implicated in a wide array of nosocomial infections, respiratory tract infections and urinary tract infections. Their *O*-antigen repeating unit oligosaccharides may be used as tools for serology or to elicit specific antibody responses using a vaccination strategy that has been successfully demonstrated for *Haemophilus influenza* type B, *Streptococcus pneumonia*, *Neisseria meningitidis*, and *Meningococcal* bacterial strains[45,46]. Several synthetic oligosaccharide constructs are now in clinical trials[47]. To the best of our knowledge, the α-(1,3)-rhamnopyranosyl-α-(1,4)-glucopyranosyl repeating unit has not been prepared before. We envisioned that oligosaccharides **28** – **30** could be assembled from rhamnosyl donor **1** and glucosyl donor **20** in a series of glycosylation and deprotection steps. In addition, incorporation of an azidopentyl linker would enable protein conjugation. C-3 acyl NGP would guarantee α-selective rhamnosylations. Glucosyl donor **20** was prepared in a six-step sequence in good yields (Supplementary Fig. 20).

First, the C-3 anisoyl-donor **1** was coupled with 5-azidopentanol (Fig. 6). The reaction using triflic acid (TfOH) and *N*-iodosuccinimide (NIS) in $CH_2Cl_2$ was chosen for its relative ease of execution and scaled

**A:**

### Rate law

$$\frac{d[OTf]}{dt} = R_{\alpha \to OTf} = k_{intra}[\alpha] + k_{inter}[\alpha][OTf]$$

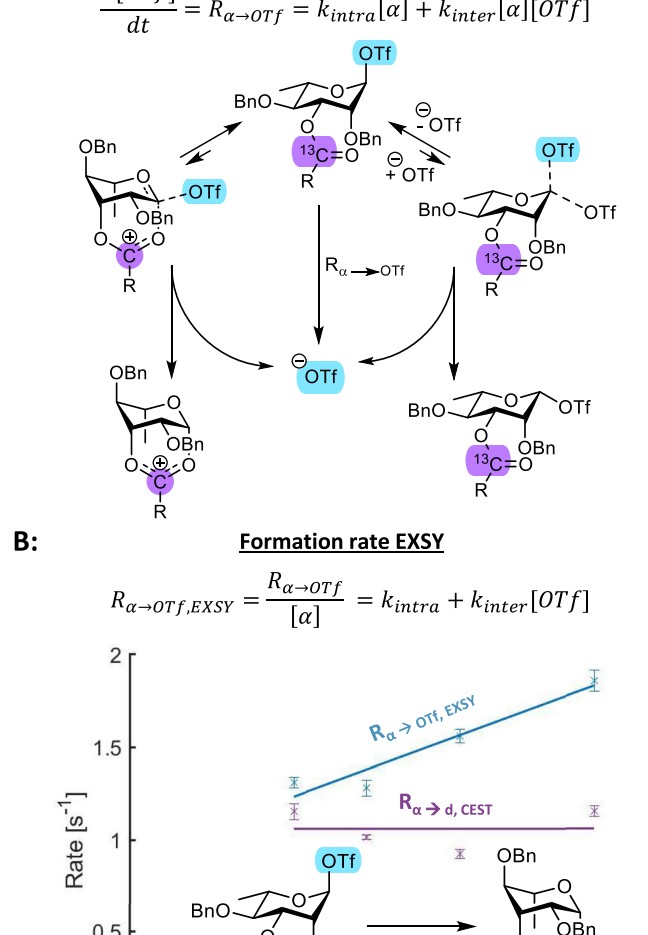

**B:**

### Formation rate EXSY

$$R_{\alpha \to OTf, EXSY} = \frac{R_{\alpha \to OTf}}{[\alpha]} = k_{intra} + k_{inter}[OTf]$$

**Fig. 5 | Reaction kinetics of the rhamnosyl donors. A)** $S_N1$- and $S_N2$-like triflate dissociation mechanisms; **B)** Reaction order determination for both triflate dissociation ($R_{\alpha \to OTf, EXSY}$) and dioxanium ion formation ($R_{\alpha \to d, CEST}$).

up. The product was directly deacylated, affording acceptor **21** in good yields over two steps. Next, the acceptor was coupled to glucosyl donor **20** to afford disaccharide **22**. Initial attempts to control anomeric selectivity with $Et_2O$ and THF as stereo-directing additives were inadequate (Supplementary Table 2). Instead, in situ imidinium adduct formation[48] was investigated using the formamides DMF[49,50], *N*-formylmorpholine (NFM) with tetrabutylammonium iodide (TBAI)[51] and methyl(phenyl)formamide (MPF)[52]. Stereoselective control with DMF-derived imidinium adducts was moderate to good, varying from 5:1 – 10:1 α:β. MPF did not provide the desired selectivity. In addition, problems in separating disaccharide **22** from MPF were encountered. Similar to the results reported by Mong et al., the NFM/TBAI method displayed excellent α-selectivity, although yields were disappointing. In contrast, reactions with DMF displayed yields up to 88%. In addition, the relatively small amounts of β-anomer could be removed via column chromatography. Accordingly, we advanced with DMF as the additive-of-choice for stereoselective glucosylations.

The 2-methylnaphthyl (Nap) ether was then removed via DDQ-mediated oxidation in moderate yields. Disaccharide acceptor **23** was subsequently coupled to rhamnosyl donor **1**, which demonstrates the utility of C-3 NGP in combination with secondary alcohol acceptors, and immediately deacylated to give trisaccharide acceptor **24** in good yield and excellent α-selectivity. The electron-donating character of the anisoyl group initially caused difficulties in deacylation. However, these were resolved after heating the mixture to 40 °C. The acceptor was again coupled to glucosyl donor **20** with DMF as additive, affording the fully protected tetrasaccharide **25**. Prior deprotection of the Nap ether in disaccharide **22** were less successful than anticipated, presumably because additional benzyl ethers had been cleaved. Notably, attempts to cleave the Nap ether with DDQ at lower temperatures or with hexafluoroisopropanol/HCl/TES failed. Therefore, the Nap ether was again removed via DDQ-mediated oxidation under standard conditions to give tetrasaccharide acceptor **26** in moderate yields. Finally, a third rhamnosylation with a consecutive deacylation step afforded the target pentasaccharide **27**. After purification with size-exclusion chromatography, NMR analysis again confirmed excellent α-selective anomeric control.

Finally, a number of failed attempts were made to hydrogenate trisaccharide **24** with palladium on carbon (Pd/C), palladium hydroxide (Pd(OH)₂) or a mix of both as earlier research suggested that this could increase the efficacy[53]. Instead, we adapted a method that was optimized for the hydrogenolysis of oligosaccharides in which Pd/C (Evonik Noblyst 10%) was first pre-treated with HCl in DMF/H₂O[54,55]. Consequently, the desired trisaccharide **28**, tetrasaccharide **29** and pentasaccharide **30** were synthesized in good yields. By conjugating them to immunogenic proteins, they may in future be used for potential vaccine development. Furthermore, the excellent stereo-directing properties of the C-3 acyl group on rhamnosides in the context of complex oligosaccharide synthesis establishes this method as a reliable alternative for the more established C-2 acyl participation. This abrogates the need for a C-2 participating group hence introducing new possibilities in the design and synthesis of oligosaccharide modified at this position.

In conclusion, we present experimental evidence of C-3 acyl NGP in ʟ-rhamnose derivatives using our optimized workflow. First, we derived the preferred reactive glycosylation intermediate in the gas-phase. IR spectra were in accordance with the theoretical DFT spectra for 1,3-bridged dioxanium ions. Next, we have presented direct proof for C-3 acyl participation in the solution-phase. CEST NMR, ¹H NMR, ¹³C NMR, COSY, HSQC, and HMBC confirmed the formation of a 1,3-bridged dioxanium ion species that is stabilized through resonance by employing a C-3 *p*-anisoyl ester. Kinetics were established of the chemical exchange between the α-triflate and the dioxanium ion and between the α-triflate and the free triflate, using ¹³C CEST NMR and ¹⁹F EXSY, respectively. These were in good accordance with each other, which is indicative of a dioxanium-dependent rather than a triflate-dependent reaction mechanism. Lastly, we applied this methodology to the stereoselective synthesis of α- ʟ-rhamnose containing oligosaccharides **28** - **30**. These results highlight the power of combining analytical techniques in elucidating reaction mechanisms and reinforce our hypothesis that α-selectivity in manno-type sugars can be attained using C-3 acyl protecting groups through the formation of 1,3-bridged dioxanium ions.

## Methods

### Ion spectroscopy in a modified ion trap mass spectrometer

IRIS experiments were performed in a quadrupole ion trap mass spectrometer (Bruker, AmaZon Speed ETD) that has been modified to provide optical access to the stored ions for spectroscopy experiments. Details of these modifications and operation of the experiment are described elsewhere[34]. Ammonium adducts of compounds **1** and **3** were generated by electrospray ionization from solutions of 10⁻⁶ M (in 1:1 Acetonitrile/water) containing 2% ammonium acetate and introduced at 2 µl min⁻¹. In order to generate the relevant oxonium products, mass-

**Fig. 6 | Synthesis of oligosaccharides 28 – 30.** α-Selective oligosaccharide synthesis using C-3 acyl NGP-assisted rhamnosylations and formamide adduct-assisted glucosylations.

selected precursor ions of interest were collisionally activated for 40 ms with an amplitude parameter of 0.2-0.4 V. The oxonium fragments are then mass isolated in an additional MS/MS stage and ultimately irradiated by the tunable FELIX mid-infrared laser beam[56]. The FEL was tuned to provide 10 μs optical pulses at 10 Hz having 30–60 mJ pulse energy over the entire tuning range (bandwidth ~0.4% of the centre frequency). Hereby the pulse energy used for experiment was attenuated accordingly to avoid saturation of the signal. Upon absorption of a sufficient number of photons unimolecular dissociation is induced and frequency-dependent fragmentation is observed by monitoring fragment ion intensities with the mass spectrometer. An infrared vibrational spectrum can be generated by relating the precursor ion intensity to the summed fragment intensities in the observed mass spectra (yield = ΣI(fragment ions)/ΣI(parent+fragment ions)) for each frequency position (3 cm⁻¹ step size). The yield is obtained from several averaged mass spectra and is linearly corrected for laser power. The IR frequency is calibrated using a grating spectrometer.

### Generation of computational IR spectra

Vibrational spectra of candidate structures were generated using a previously reported workflow[22] representations for the oxocarbenium and C-1,C-3 dioxanium ions served as the input for the cheminformatics toolbox RDKit[57]. A series of 500 random conformations were generated for each ion using the distance geometry algorithm, which were subsequently minimized using the MMFF94 classic forcefield. Based on the root-mean-squared distance between them the 40 most distinctive geometries were selected which served as an input for semiempirical PM6 minimization and vibrational analysis with Gaussian16 Rev. C.01[58]. After being filtered for duplicates, the remaining structures were minimized at the B3LYP/6-31 + + G(d,p) level, followed by vibrational analysis and a single point energy calculation at the MP2/6-31 + + G(d,p) level. Relative energies are based on the combined MP2 electronic energy and the Gibbs free energy (T = 298.15 K) from the B3LYP vibrational analysis. The harmonic vibrational line spectra were frequency scaled using a scaling factor of 0.975 and broadened using a Gaussian function with a full-width at half-maximum of 25 cm⁻¹ to resemble experimental peak widths.

### VT-NMR

Technical settings for recording CEST profiles and exchange kinetics by either ¹³C CEST NMR or ¹⁹F EXSY are displayed in the Supplementary Information (pages S7-12). The synthesis of the ¹³C enriched probes is described in the Supplementary Information. Glycosyl thioether donor (1.0 eq, typically 15 mg) and Ph₂SO (1.1 eq) were weighted and dissolved in dried DCM-$d_2$ (500 μL). Two spherical molecular sieves (4 or 5 Å) were added to the NMR tube and the tube was transferred to an analytical

scale where internal standard (either trimethyl(trifluoromethyl)silane, or trimethyl(4-trifluoromethylphenyl)silane) was added. A stock solution of Tf$_2$O was prepared in DCM-$d_2$ such that upon addition of stock solution (0.1 mL), the desired amount Tf$_2$O (1.5 eq.) could be added. The excess Tf$_2$O was added to assure full consumption of Ph$_2$SO. When the NMR sample and Tf$_2$O stock solution were ready, the NMR tube was cooled to -80 °C (dry ice/acetone bath) and to the cold tube was added the freshly prepared Tf$_2$O stock solution (100 μL). The solution generally becomes (light) yellow upon addition of Tf$_2$O, was shaken quickly (3x) and carefully transferred to the NMR. In the probe the temperature was heated to -40°C until full consumption of the initially formed species with a resonance around δ$_H$ ≈ 6.5 ppm was observed (typically 1–2 h). The sample was then cooled to −80 °C at which a battery of kinetic and characterization experiments were conducted. A 1.0 M solution of tetrabutylammonium triflate (TBAT) was prepared in DCM-d$_2$. To the solution was added activated molecular sieves (4 Å) and the solution was stored under argon at −80 °C. This solution was removed from the −80 °C fridge 60 min before the NMR experiment. NMR experiments at various concentrations triflate anion were executed as described above with respect to sample preparation. After activation at the desired temperature, the probe was set to −80 °C. At this temperature, the sample displayed an exchange (R$_{α → OTf, EXSY}$) of about 1 s$^{-1}$ to allow sufficient exchange at the lowest concentration and sufficient opportunity to increase as a consequence of the increased triflate concentration before falling out the window of EXSY NMR. After recording the triflate dissociation under standard conditions, the sample was removed from the probe, quickly stored in a dry ice/acetone bath (−80 °C) and the TBAT solution was added (20 μL). The sample was quickly shaken to homogenize the solution (3x) and was carefully transferred to the probe. The sample was locked to DCM-$d_2$, tuned, and shimmed before performing NMR experiments. After finishing the EXSY and CEST experiments, the cycle was repeated for two more time (by adding 30 μL and 50 μL TBAT solution). In the data workup, the internal standard was used to accurately correct the concentration to volume and TBAT added.

**General synthetic conditions.** Synthetic product characterisations were recorded with a Bruker 500 MHz AVANCE III spectrometer or JEOL 500 ECZ-R spectrometer. The Bruker 500 MHz Avance III spectrometer is equipped with a Prodigy BB cryoprobe. The JEOL 500 ECZ-R spectrometers were equipped with either a SuperCOOL broadband probe, ROYAL broadband probe, or ROYAL HFX broadband probe. Chemical shifts are reported in parts per million (ppm) with tetramethylsilane (TMS) as the internal standard or solvent residual signals (SRP) if stated otherwise. $^1$H NMR spectroscopic data is presented as follows: chemical shift, multiplicity (s = singlet, d = doublet, t = triplet, dd = doublet of doublets, dt = doublet of triplets, m = multiplet and/or multiple resonances), coupling constant (*J*) in hertz (Hz), integration and assignments. All NMR signals were assigned based on $^1$H NMR, $^{13}$C NMR, COSY, HSQC, HMBC, TOCSY, NOESY and ROESY experiments. Mass spectra were recorded with a JEOL JMST100CS AccuTOF mass spectrometer. Automatic silica-flash column chromatography was done with a Biotage Isolera Spektra One, using pre-packed cartridges ultrapure irregular silica gel (Screening Devices, 40-63 μm, 60 Å). Gel-filtration chromatography was performed using polyacrylamide Bio-Gel P2 beads (Bio-rad, Milli-Q as eluent) or styrene-divinylbenzene Bio-beads S-X1 resin (Bio-rad, DCM as eluent). TLC analysis was conducted on Silica gel F254 (Merck KGaA) with detection by UV absorption (254 nm) where applicable and by dipping in a stain followed by heating. Stains used for TLC analysis were either 10% sulphuric acid in MeOH, cerium molybdate stain (0.03 M (NH$_4$)$_6$Mo$_7$O$_{24}$·4H$_2$O; 6 mM Ce(NH$_4$)$_4$(SO$_4$)$_4$·2H$_2$O; 1 M H$_2$SO$_4$ in H$_2$O), potassium permanganate (0.06 M KMnO$_4$; 0.5 M, K$_2$CO$_3$; 0.02 M NaOH in H$_2$O) or ninhydrin (0.08 M ninhydrin in *n*-BuOH:AcOH, 97:3 v/v). Primary azides were stained by dipping them in 10% PPh$_3$ in DCM prior to dipping them in a ninhydrin stain. Reactions that used

anhydrous solvents were performed under Schlenk conditions and were conducted under an argon atmosphere. Molecular Sieves (0.4 nm) were activated overnight by heating *in vacuo* at 150 °C. Reactions with 2,3-dichloro-5,6-dicyano-1,4-benzoquinone (DDQ) were washed with an aqueous solution of 0.7% ascorbic acid, 1.5% citric acid and 0.9% NaOH (0.9%), abbreviated as DDQ mixture.

## Oligosaccharide synthesis
The synthesis of glycosyl donor **1** and acceptor **20** is described in the Supplementary Information (pages S26-32). TLC, NMR, and HRMS data of the synthesized compounds is reported in the Supplementary Information (pages S23-69).

**5-Azidopentyl 2,4-di-O-benzyl-α-L-rhamnopyranoside (21).** Thioglycoside **1** (3.1 g, 5.3 mmol, 1.0 eq) was dissolved in anh. DCM (75 mL). 5-Azidopentanol (0.85 mL, 6.5 mmol, 1.2 eq) was added. The solution was cooled down to 0 °C. Molecular sieves (4 Å) were added, after which the solution was stirred for 90 min. The solution was cooled down to −78 °C. NIS (1.2 g, 5.4 mmol, 1.0 eq) and TfOH (24 μL, 0.27 mmol, 0.051 eq) were added, respectively. The solution was stirred at −78 °C to 0 °C for 30 min, turning bright red over time. The solution was quenched with TEA (3.0 mL). The mixture was filtered over celite, after which the filtrate was washed with 10% aq. Na$_2$S$_2$O$_3$. The organic layer was dried with MgSO$_4$, filtered and evaporated *in vacuo*. The residue was dissolved in anh. MeOH (75 mL). 2.0 molar NaOMe in MeOH (3.0 mL, 5.3 mmol, 1.0 eq) was added. The solution was stirred at R.T. for 64 hrs. DOWEX 50 W X8(H$^+$) was added, after which the mixture was stirred for an additional 30 min. The solution was filtered and evaporated *in vacuo*. The residue was purified using silica-flash column chromatography (0 – 10% EtOAc in Tol), yielding monosaccharide **21** as a colourless oil (2.2 g, 4.9 mmol, 91%).

R$_f$ = 0.24 (EtOAc:Tol, 6:94 v/v); $^1$**H NMR** (500 MHz, CDCl$_3$): δ 7.39 – 7.26 (m, 10H, 10x Ar**H**, OBn), 4.90 (d, *J* = 11.1 Hz, 1H, PhC**H$_a$**H$_b$, 4-OBn), 4.78 (d, *J* = 1.7 Hz, 1H, **H-1**), 4.74 (d, *J* = 11.8 Hz, 1H, PhC**H$_a$**H$_b$, 2-OBn), 4.66 (d, *J* = 11.0 Hz, 1H, PhCH$_a$**H$_b$**, 4-OBn), 4.59 (d, *J* = 11.7 Hz, 1H, PhCH$_a$**H$_b$**, 2-OBn), 3.93 (td, *J* = 9.2, 3.7 Hz, 1H, **H-3**), 3.71 (dd, *J* = 3.8, 1.7 Hz, 1H, **H-2**), 3.69 – 3.61 (m, 2H, **H-5**, OC**H$_a$**H$_b$CH$_2$), 3.38 – 3.29 (m, 2H, **H-4**, OCH$_a$**H$_b$**CH$_2$), 3.26 (t, *J* = 6.9 Hz, 2H, C**H$_2$**N$_3$), 2.28 (d, *J* = 9.3 Hz, 1H, 3-O**H**), 1.63 – 1.52 (m, 4H, OCH$_2$C**H$_2$**, C**H$_2$**CH$_2$N$_3$), 1.44 – 1.37 (m, 2H, C**H$_2$**CH$_2$CH$_2$N$_3$), 1.33 (d, *J* = 6.3 Hz, 3H, 6-C**H$_3$**); $^{13}$**C NMR** (126 MHz, CDCl$_3$): δ 138.52 (ArCCH$_2$, 4-OBn), 137.78 (ArCCH$_2$, 2-OBn), [128.59, 128.44, 128.08, 128.03, 128.00, 127.77 (OBn)], 98.89 (**C–1**), 82.35 (**C-4**), 78.77 (**C-2**), 75.17 (Ph**C**H$_2$, 4-OBn), 73.08 (Ph**C**H$_2$, 2-OBn), 71.72 (**C-3**), 67.19 (O**C**H$_2$CH$_2$), 67.17 (**C-5**), 51.31 (**C**H$_2$N$_3$), 29.00 (OCH$_2$**C**H$_2$), 28.65 (**C**H$_2$CH$_2$N$_3$), 23.45 (**C**H$_2$CH$_2$CH$_2$N$_3$), 18.04 (**C-6**); **HR-ESI-TOF/MS (m/z):** [M+Na]$^+$ calcd. for C$_{25}$H$_{33}$N$_3$O$_5$Na, 478.23179; found, 478.23140.

**5-Azidopentyl [2,3,6-tri-O-benzyl-4-O-(naphthalene-2-ylmethyl)-α-D-glucopyranosyl]-(1→3)-2,4-di-O-benzyl-α-L-rhamnopyranoside (22).** Donor **20** (1.6 g, 2.3 mmol, 1.4 eq) and anh. DMF (1.9 mL, 25 mmol, 16 eq) were dissolved in anh. DCM (10 mL). The solution was cooled down to 0 °C. Molecular sieves (4 Å) were added, after which the solution was stirred for 90 min. NIS (0.50 g, 2.2 mmol, 1.4 eq) and TMSOTf (0.40 mL, 2.2 mmol, 1.4 eq) were added, respectively. The solution was stirred at 0 °C for 60 min, after which a solution of acceptor **21** (0.71 g, 1.6 mmol, 1.0 eq) in anh. DCM (10 mL) was added via a canula. The mixture was stirred at 0 °C – R.T. for 20 hrs, after which TLC showed full consumption of the acceptor. The reaction was quenched with TEA (0.50 mL) and stirred for an additional 10 min. The solution was filtered over celite and diluted with DCM. The solution was washed with 10% aq. Na$_2$S$_2$O$_3$. The organic layer was dried with MgSO$_4$, filtered and evaporated *in vacuo*. The residue was purified using silica-flash column chromatography (0 – 25% EtOAc in PE$^{100 °C-140 °C}$), yielding disaccharide **22** as a pale yellow oil (1.4 g, 1.3 mmol, 85%, 6:1 α/β).

R$_f$ = 0.46 (EtAc:Tol, 8:92 v/v); **¹H NMR** (500 MHz, CDCl₃): δ 7.83 − 7.80 (m, 1H, Ar**H**, ONap), 7.74 (d, *J* = 8.4 Hz, 1H, Ar**H**, ONap), 7.70 (dd, 1H, Ar**H**, ONap), 7.51 (s, 1H, Ar**H**, ONap), 7.48 − 7.43 (m, 2H, 2x Ar**H**), 7.37 − 7.33 (m, 2H, 2x Ar**H**, OBn), 7.32 − 7.18 (m, 21H, 21x Ar**H**, OBn), 7.12 − 7.05 (m, 3H, 3x Ar**H**, OBn), 5.20 (d, *J* = 3.5 Hz, 1H, **H-1$^{II}$**), 4.99 − 4.94 (m, 2H, PhC**H$_a$**H$_b$, 3$^{II}$-OBn; PhC**H$_a$**H$_b$, ONap), 4.91 (d, *J* = 10.5 Hz, 1H, PhC**H$_a$**H$_b$, 4$^I$-OBn), 4.88 − 4.82 (m, 2H, PhC**H$_a$**CH$_b$, 2$^I$-OBn; PhCH$_a$**H$_b$**, 3$^{II}$-OBn), 4.75 (s, 2H, PhC**H$_2$**, 2$^{II}$-OBn), 4.69 (d, *J* = 2.1 Hz, 1H, **H-1$^I$**), 4.61 − 4.53 (m, 4H, PhCH$_a$**H$_b$**, 4$^I$-OBn; PhCH$_a$**H$_b$**, ONap; PhC**H$_a$**H$_b$, 6$^{II}$-OBn), 4.27 (d, *J* = 12.1 Hz, 1H, PhCH$_a$**H$_b$**, 6$^{II}$-OBn), 4.16 (t, *J* = 9.4 Hz, 1H, **H-3$^{II}$**), 4.12 − 4.06 (m, 2H, **H-3$^I$, H-5$^{II}$**), 3.86 (t, *J* = 2.5 Hz, 1H, **H-2$^I$**), 3.80 (t, *J* = 9.5 Hz, 1H, **H-4$^{II}$**), 3.70 − 3.63 (m, 3H, **H-2$^{II}$, H-4$^I$, H-5$^I$**), 3.62 − 3.56 (m, 2H, **H-6$_a$**$^{II}$, OC**H$_a$**H$_b$CH$_2$), 3.45 (dd, *J* = 10.8, 2.1 Hz, 1H, **H-6$_b$**$^{II}$), 3.30 (dt, *J* = 9.8, 6.4 Hz, 1H, OCH$_a$**H$_b$**CH$_2$), 3.23 (t, *J* = 6.9 Hz, 2H, C**H$_2$**N$_3$), 1.60 − 1.48 (m, 4H, OCH$_2$C**H$_2$**, C**H$_2$**CH$_2$N$_3$), 1.39 − 1.31 (m, 5H, 6$^I$-C**H$_3$**, C**H$_2$**CH$_2$CH$_2$N$_3$); **¹³C NMR** (126 MHz, CDCl₃): δ 138.75 (Ar**C**CH$_2$, OBn), 138.60 (Ar**C**CH$_2$, OBn), 138.17 (Ar**C**CH$_2$, OBn), 137.98 (Ar**C**CH$_2$, OBn), 137.90 (Ar**C**CH$_2$, OBn), 136.10 (Ar**C**CH$_2$, ONap), 133.24 (Ar**C**C$_2$,ONap), 132.89 (Ar**C**C$_2$,ONap), [128.48, 128.36, 128.34, 128.29, 128.23, 128.03, 127.99, 127.89, 127.87, 127.69, 127.65, 127.64, 127.53, 127.46 (OBn; ONap)], [126.26, 125.97, 125.77 (ONap)], 98.16 (**C−1$^I$**), 95.05 (**C−1$^{II}$**), 82.28 (**C-3$^{II}$**), 80.17 (**C-4$^I$**), 79.60 (**C-2$^{II}$**), 77.86 (**C-4$^{II}$**), 76.19 (**C-3$^I$**), 75.62 (Ph**C**H$_2$, 4$^I$-OBn), 75.53 (Ph**C**H$_2$, 3$^{II}$-OBn), 75.49 (**C-2$^I$**), 75.00 (Ph**C**H$_2$, ONap), 73.36 (2 C, Ph**C**H$_2$, 2$^{II}$-OBn; Ph**C**H$_2$, 6$^{II}$-OBn), 73.29 (Ph**C**H$_2$, 2$^I$-OBn), 70.40 (**C-5$^{II}$**), 68.28 (**C-5$^I$**), 68.16 (**C-6$^{II}$**), 67.18 (O**C**H$_2$CH$_2$), 51.28 (**C**H$_2$N$_3$), 28.97 (O**C**H$_2$CH$_2$), 28.62 (**C**H$_2$CH$_2$N$_3$), 23.38 (**C**H$_2$CH$_2$CH$_2$N$_3$), 18.05 (**C-6$^I$**); **HR·ESI·TOF/MS (m/z):** [M+Na]⁺ calcd. for C$_{63}$H$_{69}$N$_3$O$_{10}$Na, 1050.48806; found, 1050.48554.

**5-Azidopentyl [2,3,6-tri-*O*-benzyl-α-ᴅ-glucopyranosyl]-(1→3)-2,4-di-*O*-benzyl-α-ʟ-rhamnopyranoside (23).** Compound **22** (2.51 g, 2.07 mmol, 1.0 eq) was dissolved in DCM:H₂O (50 mL, 9:1 v/v). DDQ (942 mg, 4.15 mmol, 1.9 eq) was added. The mixture was vigorously stirred under the exclusion of light for 3 hrs, after which DDQ mixture (5.0 mL) was added. The organic layer was extracted and washed with DDQ mixture (3 × 50 mL, aq. NaHCO₃ (sat.) (50 mL) and brine (50 mL), respectively. The organic layer was dried with MgSO₄, filtered and evaporated *in vacuo*. The residue was purified using silica-flash column chromatography (0 − 25% EtOAc in PE$^{100-140}$), yielding disaccharide **23** as a colourless oil (1.27 g, 1.43 mmol, 69.1%).

R$_f$ = 0.22 (EtAc:Tol, 8:92 v/v); **¹H NMR** (500 MHz, CDCl₃): δ 7.36 − 7.22 (m, 25H, 25x Ar**H**), 5.18 (d, *J* = 3.5 Hz, 1H, **H-1$^{II}$**), 4.94 (d, *J* = 11.4 Hz, 1H, PhC**H$_a$**H$_b$, 3$^{II}$-OBn), 4.88 (d, *J* = 10.9 Hz, 1H, PhC**H$_a$**H$_b$, 4$^I$-OBn), 4.82 (d, *J* = 11.7 Hz, 1H, PhC**H$_a$**CH$_b$, 2$^I$-OBn), 4.75 (d, *J* = 11.6 Hz, 1H, PhCH$_a$**H$_b$**, 3$^{II}$-OBn), 4.73 − 4.69 (m, 3H, **H-1$^I$**, PhC**H$_2$**, 2$^{II}$-OBn), 4.62 − 4.58 (m, 2H, PhCH$_a$C**H$_b$**, 2$^I$-OBn; PhCH$_a$**H$_b$**, 4$^I$-OBn), 4.50 (d, *J* = 12.1 Hz, 1H, PhC**H$_a$**H$_b$, 6$^{II}$-OBn), 4.39 (d, *J* = 12.1 Hz, 1H, PhCH$_a$**H$_b$**, 6$^{II}$-OBn), 4.10 (dd, *J* = 8.7, 2.9 Hz, 1H, **H-3$^I$**), 3.99 (dt, *J* = 10.0, 3.7 Hz, 1H, **H-5$^{II}$**), 3.90 (t, *J* = 9.3 Hz, 1H, **H-3$^{II}$**), 3.86 (t, *J* = 2.6 Hz, 1H, **H-2$^I$**), 3.71 − 3.57 (m, 5H, **H-2$^{II}$, H-4$^I$, H-4$^{II}$, H-5$^I$**, OC**H$_a$**H$_b$CH$_2$), 3.53 (qd, *J* = 10.7, 3.9 Hz, 2H, 2x H6$^{II}$), 3.30 (dt, *J* = 10.0, 6.4 Hz, 1H, OCH$_a$**H$_b$**CH$_2$), 3.21 (t, *J* = 6.9 Hz, 2H, C**H$_2$**N$_3$), 2.11 (d, *J* = 3.0 Hz, 1H, 4$^{II}$-O**H**), 1.59 − 1.47 (m, 4H, OCH$_2$C**H$_2$**, C**H$_2$**CH$_2$N$_3$), 1.39 − 1.29 (m, 5H, 6$^I$-C**H$_3$**, C**H$_2$**CH$_2$CH$_2$N$_3$); **¹³C NMR** (126 MHz, CDCl₃): δ 138.73 (Ar**C**CH$_2$, OBn), 138.55 (Ar**C**CH$_2$, OBn), 138.21 (Ar**C**CH$_2$, OBn), 138.07 (Ar**C**CH$_2$-, OBn), [128.53, 128.38, 128.35, 128.28, 128.26, 128.25, 128.04, 127.94, 127.79, 127.73, 127.70, 127.68, 127.61, 127.54, 127.49 (OBn)], 98.12 (**C-1$^I$**), 95.05 (**C-1$^{II}$**), 81.33 (**C-3$^{II}$**), 80.13 (**H-4$^I$**), 79.34 (**C-2$^{II}$**), 76.18 (**C-3$^I$**), 75.57 (**C-2$^I$**), 75.25 (Ph**C**H$_2$, 4$^I$-OBn), 75.15 (Ph**C**H$_2$, 3$^{II}$-OBn), 73.39 (Ph**C**H$_2$, 6$^{II}$-OBn), 73.25 (Ph**C**H$_2$, 2$^I$-OBn), 73.08 (Ph**C**H$_2$, 2$^{II}$-OBn), 71.08 (**C-4$^{II}$**), 70.17 (**C-5$^{II}$**), 69.33 (**C-6$^{II}$**), 68.27 (C-5$^I$), 67.20 (O**C**H$_2$CH$_2$), 51.26 (**C**H$_2$N$_3$), 28.95 (O**C**H$_2$CH$_2$), 28.60 (**C**H$_2$CH$_2$N$_3$), 23.37 (**C**H$_2$CH$_2$CH$_2$N$_3$), 18.05 (**C-6$^I$**); **HR·ESI·TOF/MS (m/z):** [M+Na]⁺ calcd. for C$_{52}$H$_{61}$N$_3$O$_{10}$Na, 910.42546; found, 910.42731.

**5-Azidopentyl [2,4-di-*O*-benzyl-α-ʟ-rhamnopyranosyl]-(1→4)-[2,3,6-tri-*O*-benzyl-α-ᴅ-glucopyranosyl]-(1→3)-2,4-di-*O*-benzyl-α-ʟ-rhamnopyranoside (24).** Donor **1** (1.04 g, 1.82 mmol, 1.4 eq) and acceptor **23** (1.13 g, 1.28 mmol, 1.0 eq) were dissolved in anh. DCM (30 mL). The solution was cooled down to 0 °C. Molecular sieves (4 Å) were added, after which the solution was stirred for 90 min. The solution was cooled down to -78 °C. NIS (402 mg, 1.79 mmol, 1.4 eq) and TfOH (7.0 μL, 79 μmol, 0.062 eq) were added, respectively. The solution was stirred at -78 °C to 0 °C for 60 min, turning bright red over time. The solution was quenched with TEA (0.50 mL) and stirred for an additional 10 min. The mixture was filtered over celite, after which the filtrate was washed with 10% aq. Na₂S₂O₃. The organic layer was dried with MgSO₄, filtered and evaporated *in vacuo*. The residue was dissolved in MeOH:THF (50 mL, 3:2 v/v). 5.4 molar NaOMe in MeOH (0.50 mL, 2.7 mmol, 2.1 eq) was added. The solution was stirred for 64 hrs. The solution was heated to 40 °C, after which it was stirred for an additional 20 hrs. DOWEX 50 W X8(H⁺) was added, after which the mixture was stirred for an additional 15 min. The solution was filtered and evaporated *in vacuo*. The residue was purified using silica-flash column chromatography (0 − 8% EtOAc in Tol), yielding trisaccharide **24** as a colourless oil (1.36 g, 1.12 mmol, 87.5%).

R$_f$ = 0.24 (EtAc:Tol, 8:92 v/v); **¹H NMR** (500 MHz, CDCl₃): δ 7.38 − 7.17 (m, 35H, 35x Ar**H**, OBn), 5.15 (d, *J* = 3.5 Hz, 1H, **H-1$^{II}$**), 5.04 (d, *J* = 1.7 Hz, 1H, **H-1$^{III}$**), 5.00 (d, *J* = 10.6 Hz, 1H, PhC**H$_a$**H$_b$, 3$^{II}$-OBn), 4.90 (d, *J* = 11.0 Hz, 1H, PhC**H$_a$**H$_b$, 4$^I$-OBn), 4.85 − 4.80 (m, 2H, PhC**H$_a$**H$_b$, 3$^{III}$-OBn; PhC**H$_a$**H$_b$, 2$^I$-OBn), 4.77 (d, *J* = 10.6 Hz, 1H, PhCH$_a$**H$_b$**, 3$^{II}$-OBn), 4.72 − 4.68 (m, 3H, **H-1$^I$**, PhC**H$_2$**, 2$^{II}$-OBn), 4.63 (d, *J* = 10.9 Hz, 1H, PhCH$_a$**H$_b$**, 4$^I$-OBn), 4.60 − 4.54 (m, 2H, PhCH$_a$**H$_b$**, 2$^I$-OBn; PhCH$_a$**H$_b$**, 4$^{III}$-OBn), 4.50 − 4.45 (m, 2H, PhC**H$_a$**H$_b$, 2$^{III}$-OBn; PhC**H$_a$**H$_b$, 6$^{II}$-OBn), 4.33 (d, *J* = 12.0 Hz, 1H, PhCH$_a$**H$_b$**, 6$^{II}$-OBn), 4.25 (d, *J* = 11.8 Hz, 1H, PhCH$_a$**H$_b$**, 2$^{III}$-OBn), 4.08 (dd, *J* = 8.9, 2.9 Hz, 1H, **H-3$^I$**), 3.98 − 3.93 (m, 2H, **H-3$^{II}$, H-5$^{II}$**), 3.92 − 3.82 (m, 4H, **H-2$^I$, H-4$^{II}$, H-3$^{III}$, H-5$^{III}$**), 3.72 − 3.62 (m, 3H, **H-2$^{II}$, H-4$^I$, H-5$^I$**), 3.62 − 3.57 (m, 1H, OC**H$_a$**H$_b$CH$_2$), 3.49 (dd, *J* = 3.7, 1.6 Hz, 1H, **H-2$^{III}$**), 3.45 (dd, *J* = 11.4, 1.9 Hz, 1H, **H-6$_a$**$^{II}$), 3.36 − 3.28 (m, 2H, **H-6$_b$**$^{II}$, OCH$_a$**H$_b$**CH$_2$), 3.27 − 3.21 (m, 3H, **H-4$^{III}$**, C**H$_2$**N$_3$), 2.23 (d, *J* = 8.9 Hz, 1H, 3$^{III}$-O**H**), 1.60 − 1.49 (m, 4H, C**H$_2$**CH$_2$N$_3$, OCH$_2$C**H$_2$**), 1.41 − 1.33 (m, 5H, 6$^I$-C**H$_3$**, C**H$_2$**CH$_2$CH$_2$N$_3$), 0.98 (d, *J* = 6.2 Hz, 3H, 6$^{III}$-C**H$_3$**); **¹³C NMR** (126 MHz, CDCl₃): δ 138.84 (Ar**C**CH$_2$, OBn), 138.53 (Ar**C**CH$_2$, OBn), 138.46 (Ar**C**CH$_2$, OBn), 138.21 (Ar**C**CH$_2$, OBn), 137.89 (Ar**C**CH$_2$, OBn), 137.83 (Ar**C**CH$_2$, OBn), [128.50, 128.48, 128.36, 128.31, 128.29, 128.22, 128.16, 128.04, 128.00, 127.87, 127.81, 127.77, 127.73, 127.71, 127.69, 127.60, 127.56, 127.52, 127.49, 127.48, 127.32 (OBn)], 98.02 (**C-1$^I$**), 97.46 (**C-1$^{III}$**), 94.45 (**C-1$^{II}$**), 82.36 (**C-4$^{III}$**), 80.28 (**C-3$^{II}$**), 80.08 (**C-2$^{II}$**), 79.95 (**C-4$^I$**), 78.43 (**C-2$^{III}$**), 76.06 (**C-3$^I$**), 75.54 (**C-2$^I$**), 75.38 (Ph**C**H$_2$, 4$^{II}$-OBn), 75.33 (Ph**C**H$_2$, 3$^{II}$-OBn), 75.10 (**C-4$^{II}$**), 74.85 (Ph**C**H$_2$, 4$^{III}$-OBn), 73.43 (Ph**C**H$_2$, 6$^{II}$-OBn), 73.36 (Ph**C**H$_2$, 2$^{II}$-OBn), 73.30 (Ph**C**H$_2$, 2$^I$-OBn), 72.51 (Ph**C**H$_2$, 2$^{III}$-OBn), 71.44 (**C-3$^{III}$**), 70.38 (**C-5$^{II}$**), 68.84 (**C-6$^{II}$**), 68.35 (**C-5$^I$**), 67.94 (**C-5$^{III}$**), 67.21 (O**C**H$_2$CH$_2$), 51.28 (**C**H$_2$N$_3$), 28.96 (O**C**H$_2$CH$_2$), 28.62 (**C**H$_2$CH$_2$N$_3$), 23.38 (**C**H$_2$CH$_2$CH$_2$N$_3$), 18.00 (**C-6$^I$**), 17.87 (**C-6$^{III}$**); **HR·ESI·TOF/MS (m/z):** [M+Na]⁺ calcd. for C$_{72}$H$_{83}$N$_3$O$_{14}$Na, 1236.57727; found, 1236.57710.

**5-Azidopentyl [2,3,6-tri-*O*-benzyl-4-*O*-(naphthalene-2-ylmethyl)-α-ᴅ-glucopyranosyl]-(1→3)-[2,4-di-*O*-benzyl-α-ʟ-rhamnopyranosyl]-(1→4)-[2,3,6-tri-*O*-benzyl-α-ᴅ-glucopyranosyl]-(1→3)-2,4-di-*O*-ben-zyl-α-ʟ-rhamnopyranoside (25).** Donor **20** (0.64 g, 0.92 mmol, 1.3 eq) and anh. DMF (0.80 mL, 10 mmol, 15 eq) were dissolved in anh. DCM (8.0 mL). The solution was cooled down to 0 °C. Molecular sieves (4 Å) were added, after which the solution was stirred for 90 min. NIS (0.21 g, 0.94 mmol, 1.4 eq) and TMSOTf (0.17 mL, 0.94 mmol, 1.4 eq) were added, respectively. The solution was stirred at 0 °C for 60 min, after which a solution of acceptor **24** (0.84 g, 0.69 mmol, 1.0 eq) in anh. DCM (4.0 mL) was added via a canula. The mixture was stirred at 0 °C – R.T. for 20 hrs, after which TLC showed full consumption of the acceptor. The reaction was quenched with TEA (0.30 mL) and stirred for an additional 10 min. The solution was filtered over celite and

diluted with DCM. The solution was washed with 10% aq. $Na_2S_2O_3$. The organic layer was dried with $MgSO_4$, filtered and evaporated *in vacuo*. The residue was purified using size-exclusion chromatography (Biorad S-X1 support) and silica-flash column chromatography (0 – 20% EtAc in $PE^{100\,°C-140\,°C}$), yielding tetrasaccharide **25** as a colourless oil (1.1 g, 0.60 mmol, 88%, 10:1 α/β).

$R_f$ = 0.46 (EtAc:Tol, 10:90 v/v); **$^1$H NMR** (500 MHz, $CDCl_3$): δ 7.83 (dd, *J* = 6.1, 3.4 Hz, 1H, Ar***H***, ONap), 7.74 (d, *J* = 8.4 Hz, 1H, Ar***H***, ONap), 7.71 (dd, *J* = 6.1, 3.3 Hz, Ar***H***, ONap), 7.51 – 7.43 (m, 3H, 3x Ar***H***, ONap), 7.39 – 7.03 (m, 51H, Ar***H***, ONap; 50x Ar***H***, OBn), 5.16 (d, *J* = 3.6 Hz, 1H, ***H-1$^{II}$***), 5.10 (d, *J* = 3.7 Hz, 1H, ***H-4$^{IV}$***), 5.09 (d, *J* = 2.2 Hz, 1H, ***H-1$^{III}$***), 4.96 – 4.90 (m, 3H, PhC***H$_a$***H$_b$, 3$^{IV}$-OBn; PhC***H$_a$***H$_b$, 4$^{I}$-OBn; PhC***H$_a$***H$_b$, ONap), 4.90 – 4.78 (m, 5H, PhC***H$_a$***H$_b$, 2$^{I}$-OBn; PhC***H$_2$***, 3$^{II}$-OBn; PhC***H$_a$***H$_b$, 3$^{IV}$-OBn; PhC***H$_a$***H$_b$, 4$^{III}$-OBn), 4.72 – 4.66 (m, 3H, ***H-1$^{I}$***, PhC***H$_2$***, 2$^{II}$-OBn), 4.66 – 4.51 (m, 7H, PhC***H$_a$***H$_b$, 2$^{I}$-OBn; PhC***H$_a$***H$_b$, 2$^{III}$-OBn; PhC***H$_2$***, 2$^{IV}$-OBn; PhC***H$_a$***H$_b$, 4$^{I}$-OBn; PhC***H$_a$***H$_b$, 4$^{III}$-OBn; PhC***H$_a$***H$_b$***, ONap), 4.44 – 4.37 (m, 3H, PhC***H$_a$***H$_b$, 2$^{III}$-OBn; PhC***H$_a$***H$_b$, 6$^{II}$-OBn; PhC***H$_a$***H$_b$, 6$^{IV}$-OBn), 4.32 (d, *J* = 11.9 Hz, 1H, PhC***H$_a$***H$_b$, 6$^{II}$-OBn), 4.19 – 4.10 (m, 3H, ***H-3$^{III}$***, ***H-3$^{I}$***, PhC***H$_a$***H$_b$, 6$^{IV}$-OBn), 4.09 – 4.03 (m, 2H, ***H-3$^{I}$***, ***H-5$^{III}$***), 4.02 – 3.90 (m, 4H, ***H-3$^{II}$***, ***H-4$^{II}$***, ***H-5$^{II}$***, ***H-5$^{III}$***), 3.88 (t, *J* = 2.6 Hz, 1H, ***H-2$^{I}$***), 3.83 – 3.74 (m, 2H, ***H-2$^{III}$***, ***H-4$^{IV}$***), 3.71 – 3.56 (m, 6H, ***H-2$^{II}$***, ***H-2$^{IV}$***, ***H-4$^{I}$***, ***H-4$^{III}$***, ***H-5$^{I}$***, OC***H$_a$***H$_b$CH$_2$), 3.46 – 3.35 (m, 3H, ***H-6$_a^{I}$***, ***H-6$_b^{II}$***, ***H-6$_a^{I}$***), 3.29 (dt, *J* = 10.0, 6.4 Hz, 1H, OCH$_a$***H$_b$***CH$_2$), 3.26 – 3.18 (m, 3H, ***H-6$_b^{IV}$***, C***H$_2$***N$_3$), 1.58 – 1.47 (m, 4H, C***H$_2$***CH$_2$N$_3$, OCH$_2$C***H$_2$***), 1.39 – 1.28 (m, 5H, 6$^{I}$-C***H$_3$***, C***H$_2$***CH$_2$CH$_2$N$_3$), 1.14 (d, *J* = 6.1 Hz, 3H, 6$^{III}$-C***H$_3$***); **$^{13}$C NMR** (126 MHz, $CDCl_3$): δ 138.70 (Ar***C***CH$_2$, OBn), 138.51 (Ar***C***CH$_2$, OBn), 138.47 (Ar***C***CH$_2$, OBn), 138.34 (Ar***C***CH$_2$, OBn), 138.23 (Ar***C***CH$_2$, OBn), 138.22 (Ar***C***CH$_2$, OBn), 138.13 (Ar***C***CH$_2$, OBn), 138.00 (Ar***C***CH$_2$, OBn), 137.77 (Ar***C***CH$_2$, OBn), 136.14 (Ar***C***CH$_2$, ONap), 133.18 (Ar***C***C$_2$, ONap), 132.83 (Ar***C***C$_2$, ONap), 128.47 (OBn), [128.32, 128.31, 128.28, 128.24, 128.21, 128.19, 128.12, 128.09, 128.06, 127.95, 127.87, 127.81, 127.75, 127.64, 127.53, 127.49, 127.39, 127.27, 127.20 (OBn; ONap)], [126.22 126.02, 125.94, 125.73 (ONap)], 98.53 (***C-1$^{III}$***), 98.07 (***C-1$^{I}$***), 95.51 (***C-1$^{II}$***), 94.07 (***C-1$^{IV}$***), 82.23 (***C-3$^{IV}$***), 80.76 (***C-3$^{II}$***), 80.19 (***C-4$^{I}$***), 79.99 (***C-2$^{II}$***), 79.76 (***C-4$^{III}$***), 79.55 (***C-2$^{IV}$***), 77.74 (***C-4$^{IV}$***), 76.93 (***C-3$^{I}$***), 75.83 (***C-2$^{I}$***), 75.58 (Ph***C***H$_2$-, 3$^{II}$-OBn), 75.53 (Ph***C***H$_2$, 4$^{III}$-OBn), 75.49 (Ph***C***H$_2$, 3$^{IV}$-OBn), 75.35 (***C-3$^{III}$***), 75.24 (Ph***C***H$_2$, 4$^{I}$-OBn), 75.05 (***C-2$^{III}$***), 74.91 (Ph***C***H$_2$, ONap), 74.60 (***C-4$^{II}$***), 73.24 (2 C, Ph***C***H$_2$, 2$^{II}$-OBn; Ph***C***H$_2$, 6$^{IV}$-OBn), 73.18 (Ph***C***H$_2$, 2$^{I}$-OBn), 73.15 (Ph***C***H$_2$, 6$^{II}$-OBn), 73.07 (Ph***C***H$_2$, 2$^{IV}$-OBn), 72.84 (Ph***C***H$_2$, 2$^{III}$-OBn), 70.68 (***C-5$^{II}$***), 70.14 (***C-5$^{IV}$***), 69.05 (***C-5$^{III}$***), 68.82 (***C-6$^{II}$***), 68.20 (***C-5$^{I}$***), 67.94 (***C-6$^{IV}$***), 67.19 (O***C***H$_2$CH$_2$), 51.21 (***C***H$_2$N$_3$), 28.95 (O***C***H$_2$CH$_2$), 28.58 (***C***H$_2$CH$_2$N$_3$), 23.32 (***C***H$_2$CH$_2$CH$_2$N$_3$), 18.05 (***C-6$^{I}$***), 18.02 (***C-6$^{III}$***); **HR-ESI-TOF/MS (m/z):** [M+Na]$^+$ calcd. for $C_{110}H_{119}N_3O_{19}Na$, 1808.83354; found, 1808.83038.

### 5-Azidopentyl [2,3,6-tri-*O*-benzyl-α-ᴅ-glucopyranosyl]-(1→3)-[2,4-di-*O*-benzyl-α-ʟ-rhamnopyranosyl]-(1→4)-[2,3,6-tri-*O*-benzyl-α-ᴅ-glucopyranosyl]-(1→3)-2,4-di-*O*-benzyl-α-ʟ-rhamnopyranoside (26).

Compound **25** (0.24 g, 0.14 mmol, 1.0 eq) was dissolved in DCM:H$_2$O (5.0 mL, 9:1 v/v). The solution was cooled down to 12 °C. DDQ (38 mg, 0.17 mmol, 1.2 eq) was added. The mixture was vigourously stirred under exclusion of light for 2 hrs, after which DCM (5.0 mL) and DDQ mixture (10 mL) were added. The organic layer was extracted and washed with H$_2$O (10 mL) and aq. NaHCO$_3$ (sat.) (10 mL), respectively. The organic layer was dried with MgSO$_4$, filtered and evaporated *in vacuo*. The residue was purified using silica-flash column chromatography (0 – 10% EtAc in Tol), yielding tetrasaccharide **26** as a colourless oil (0.15 g, 91 μmol, 67%).

$R_f$ = 0.36 (EtAc:Tol, 10:90 v/v); **$^1$H NMR** (500 MHz, $CDCl_3$): δ 7.41 – 7.10 (m, 50H, 50x Ar***H***, OBn), 5.16 (d, *J* = 3.5 Hz, 1H, ***H-1$^{I}$***), 5.10 (d, *J* = 2.1 Hz, 1H, ***H-1$^{III}$***), 5.08 (d, *J* = 3.5 Hz, 1H, ***H-1$^{IV}$***), 4.93 – 4.83 (m, 4H, PhC***H$_a$***H$_b$, 3$^{II}$-OBn; PhC***H$_a$***H$_b$, 3$^{IV}$-OBn; PhC***H$_a$***H$_b$, 4$^{I}$-OBn; PhC***H$_a$***H$_b$, 4$^{III}$-OBn), 4.83 – 4.78 (m, 2H, PhC***H$_a$***H$_b$, 2$^{I}$-OBn; PhC***H$_a$***H$_b$, 3$^{II}$-OBn), 4.74 – 4.67 (m, 4H, ***H-1$^{I}$***, PhC***H$_2$***, 2$^{II}$-OBn; PhC***H$_a$***H$_b$, 3$^{IV}$-OBn), 4.63 – 4.53 (m, 6H, PhC***H$_a$***H$_b$, 2$^{I}$-OBn; PhC***H$_a$***H$_b$, 2$^{III}$-OBn; PhC***H$_2$***, 2$^{IV}$-OBn; PhC***H$_a$***H$_b$, 4$^{I}$-OBn; PhC***H$_a$***H$_b$, 4$^{III}$-OBn; PhC***H$_a$***H$_b$, 6$^{II}$-OBn), 4.41 – 4.34 (m, 3H, PhC***H$_a$***H$_b$, 2$^{III}$-OBn; PhC***H$_a$***H$_b$, 6$^{II}$-OBn; PhC***H$_a$***H$_b$, 6$^{IV}$-OBn), 4.32 (d, *J* = 12.1 Hz, 1H, PhC***H$_a$***H$_b$, 6$^{II}$-OBn), 4.29 (d, *J* = 12.2 Hz, 1H, PhC***H$_a$***H$_b$, 6$^{IV}$-OBn), 4.11 (dd, *J* = 9.2, 2.8 Hz, 1H, ***H-3$^{III}$***), 4.07 (dd, *J* = 8.6, 3.0 Hz, 1H, ***H-3$^{I}$***), 4.02 – 3.93 (m, 4H, ***H-3$^{II}$***, ***H-5$^{II}$***, ***H-5$^{III}$***, ***H-5$^{III}$***), 3.93 – 3.83 (m, 3H, ***H-2$^{I}$***, ***H-3$^{IV}$***, ***H-4$^{II}$***), 3.77 (t, *J* = 2.5 Hz, 1H, ***H-2$^{III}$***), 3.71 – 3.57 (m, 6H, ***H-2$^{II}$***, ***H-4$^{I}$***, ***H-4$^{III}$*** ***H-4$^{IV}$***, ***H-5$^{I}$***, OC***H$_a$***H$_b$CH$_2$), 3.55 (dd, *J* = 9.6, 3.4 Hz, 1H, ***H-2$^{IV}$***), 3.45 – 3.33 (m, 4H, ***H-6$_a^{II}$***, ***H-6$_b^{II}$***, ***H-6$_a^{IV}$***, ***H-6$_b^{IV}$***), 3.30 (dt, *J* = 10.0, 6.5 Hz, 1H, OCH$_a$***H$_b$***CH$_2$), 3.21 (t, *J* = 6.9 Hz, 2H, C***H$_2$***N$_3$), 2.04 (d, *J* = 3.2 Hz, 1H, 4$^{IV}$-O***H***), 1.59 – 1.47 (m, 4H, C***H$_2$***CH$_2$N$_3$, OCH$_2$C***H$_2$***), 1.39 – 1.30 (m, 5H, 6$^{I}$-C***H$_3$***, C***H$_2$***CH$_2$CH$_2$N$_3$), 1.12 (d, *J* = 6.1 Hz, 3H, 6$^{III}$-C***H$_3$***); **$^{13}$C NMR** (126 MHz, $CDCl_3$): δ 138.70 (Ar***C***CH$_2$, OBn), 138.48 (Ar***C***CH$_2$, OBn), 138.44 (Ar***C***CH$_2$, OBn), 138.34 (Ar***C***CH$_2$, OBn), 138.24 (Ar***C***CH$_2$, OBn), 138.17 (Ar***C***CH$_2$, OBn), 138.05 (Ar***C***CH$_2$, OBn), 137.99 (Ar***C***CH$_2$, OBn), 137.88 (Ar***C***CH$_2$, OBn), [128.47, 128.45, 128.31, 128.30, 128.27, 128.24, 128.21, 128.19, 128.14, 128.06, 128.02, 127.95, 127.93, 127.74, 127.71, 127.67, 127.64, 127.63, 127.59, 127.56, 127.49, 127.42, 127.38, 127.31, 127.16 (OBn)], 98.43 (***C-1$^{III}$***), 98.06 (***C-1$^{I}$***), 95.50 (***C-1$^{II}$***), 93.76 (***C-1$^{IV}$***), 81.28 (***C-3$^{IV}$***), 80.80 (***C-3$^{II}$***), 80.17 (***C-4$^{I}$***), 79.97 (***C-2$^{II}$***), 79.70 (***C-4$^{III}$***), 79.19 (***C-2$^{IV}$***), 76.91 (***C-3$^{I}$***), 75.82 (***C-2$^{I}$***), 75.55 (Ph***C***H$_2$, 3$^{II}$-OBn), 75.22 (Ph***C***H$_2$, 4$^{I}$-OBn), 75.11 (Ph***C***H$_2$, 3$^{IV}$-OBn; Ph***C***H$_2$, 4$^{III}$-OBn), 75.07 (***C-3$^{III}$***), 74.81 (***C-2$^{III}$***), 74.48 (***C-4$^{II}$***), 73.35 (Ph***C***H$_2$, 6$^{IV}$-OBn), 73.28 (Ph***C***H$_2$, 2$^{II}$-OBn), 73.18 (Ph***C***H$_2$, 2$^{I}$-OBn), 73.11 (Ph***C***H$_2$, 6$^{II}$-OBn), 72.82 (Ph***C***H$_2$, 2$^{IV}$-OBn), 72.77 (Ph***C***H$_2$, 2$^{III}$-OBn), 71.30 (***C-4$^{IV}$***), 70.65 (***C-5$^{II}$***), 69.72 (***C-5$^{IV}$***), 69.35 (***C-6$^{IV}$***), 69.00 (***C-5$^{III}$***), 68.81 (***C-6$^{II}$***), 68.19 (***C-5$^{I}$***), 67.19 (O***C***H$_2$CH$_2$), 51.21 (***C***H$_2$N$_3$), 28.95 (O***C***H$_2$CH$_2$), 28.58 (***C***H$_2$CH$_2$N$_3$), 23.33 (***C***H$_2$CH$_2$CH$_2$N$_3$), 18.03 (***C-6$^{I}$***), 18.01 (***C-6$^{III}$***); **HR-ESI-TOF/MS (m/z):** [M+Na]$^+$ calcd. for $^{12}C_{98}{}^{13}C_1H_{111}N_3O_{19}Na$, 1669.77430; found, 1669.77732.

### 5-Azidopentyl [2,4-di-*O*-benzyl-α-ʟ-rhamnopyranosyl]-(1→4)-[2,3,6-tri-*O*-benzyl-α-ᴅ-glucopyranosyl]-(1→3)-[2,4-di-*O*-benzyl-α-ʟ-rhamnopyranosyl]-(1→4)-[2,3,6-tri-*O*-benzyl-α-ᴅ-glucopyranosyl]-(1→3)-2,4-di-*O*-benzyl-α-ʟ-rhamnopyranoside (27).

Donor **1** (62 mg, 0.11 mmol, 2.1 eq) and acceptor **26** (85 mg, 52 μmol, 1.0 eq) were dissolved in anh. DCM (3.0 mL). The solution was cooled down to 0 °C. Molecular sieves (4 Å) were added, after which the solution was stirred for 90 min. The solution was cooled down to −78 °C. NIS (25 mg, 0.11 mmol, 2.2 eq) and TfOH (1.0 μL, 11 μmol, 0.22 eq) were added, respectively. The solution was stirred at −78 °C for 15 min, after which it was stirred at -78 °C to 0 °C for 30 min, turning bright red over time. The solution was quenched with TEA (0.20 mL) and stirred for an additional 10 min. The mixture was filtered over celite, after which the filtrate was washed with 10% aq. Na$_2$S$_2$O$_3$ (10 mL). The organic layer was dried with MgSO$_4$, filtered and evaporated *in vacuo*. The residue was dissolved in MeOH:THF (5.0 mL, 3:2 v/v). 5.4 molar NaOMe in MeOH (0.10 mL, 0.54 mmol, 10 eq) was added. The solution was heated to 45 °C and stirred for 20 hrs. DOWEX 50 W X8(H$^+$) was added, after which the mixture was stirred for an additional 15 min. The solution was filtered and evaporated *in vacuo*. The residue was purified using size-exclusion chromatography (Biorad S-X1 support), yielding pentasaccharide **27** as a colourless oil (74 mg, 37 μmol, 73%).

$R_f$ = 0.35 (EtAc:Tol, 10:90 v/v); **$^1$H NMR** (500 MHz, $CDCl_3$): δ 7.44 – 7.07 (m, 60H, 60x Ar***H***, OBn), 5.17 (d, *J* = 3.6 Hz, 1H), 5.11 (s, 1H, ***H-1$^{III}$***), 5.05 (d, *J* = 3.6 Hz, 1H, ***H-1$^{IV}$***), 5.00 (s, 1H, ***H-1$^{V}$***), 4.97 (d, *J* = 10.7 Hz, 1H, PhC***H$_a$***H$_b$, 3$^{IV}$-OBn), 4.94 (d, *J* = 10.8 Hz, 1H, PhC***H$_a$***H$_b$, 4$^{I}$-OBn), 4.89 – 4.78 (m, 5H, PhC***H$_a$***H$_b$, 2$^{I}$-OBn; PhC***H$_2$***, 3$^{II}$-OBn; PhC***H$_a$***H$_b$, 4$^{III}$-OBn; PhC***H$_a$***H$_b$, 4$^{V}$-OBn), 4.75 – 4.65 (m, 4H, ***H-1$^{I}$***, PhC***H$_2$***, 2$^{II}$-OBn; PhC***H$_a$***H$_b$, 3$^{IV}$-OBn), 4.63 – 4.53 (m, 7H, PhC***H$_a$***H$_b$, 2$^{I}$-OBn; PhC***H$_a$***H$_b$, 2$^{III}$-OBn; PhC***H$_2$***, 2$^{IV}$-OBn; PhC***H$_a$***H$_b$, 4$^{I}$-OBn; PhC***H$_a$***H$_b$, 4$^{III}$-OBn; PhC***H$_a$***H$_b$, 4$^{V}$-OBn), 4.46 (d, *J* = 11.8 Hz, 1H, PhC***H$_a$***H$_b$, 2$^{V}$-OBn), 4.41 (d, *J* = 12.0 Hz, 1H, PhC***H$_a$***H$_b$, 6$^{II}$-OBn), 4.37 – 4.31 (m, 3H, PhC***H$_a$***H$_b$, 2$^{III}$-OBn; PhC***H$_a$***H$_b$, 6$^{II}$-OBn; PhC***H$_a$***H$_b$, 6$^{IV}$-OBn), 4.29 – 4.20 (m, 2H, PhC***H$_a$***H$_b$, 2$^{V}$-OBn; PhC***H$_a$***H$_b$, 6$^{IV}$-OBn), 4.13 – 4.06 (m, 2H, ***H-3$^{I}$***, ***H-3$^{III}$***), 4.02 (dd, *J* = 9.4, 6.0 Hz, 1H, ***H-5$^{III}$***), 4.00 – 3.94 (m, 2H, ***H-3$^{II}$***, ***H-5$^{I}$***), 3.93 – 3.78 (m, 7H, ***H-2$^{I}$***, ***H-3$^{IV}$***, ***H-3$^{V}$***, ***H-4$^{II}$***, ***H-4$^{IV}$***, ***H-5$^{IV}$***, ***H-5$^{V}$***), 3.75 (d, *J* = 2.7 Hz, 1H, ***H-2$^{V}$***), 3.72 – 3.55 (m, 6H, ***H-2$^{II}$***, ***H-2$^{IV}$***, ***H-4$^{I}$***, ***H-4$^{III}$***, ***H-5$^{I}$***, OC***H$_a$***H$_b$CH$_2$), 3.49 – 3.35

(m, 3H, **H-2$^V$**, **H-6$_a$$^{II}$**, **H-6$_b$$^{II}$**), 3.34 – 3.26 (m, 2H, **H-6$_a$$^{IV}$**, OCH$_a$**H$_b$**CH$_2$), 3.26 – 3.18 (m, 3H, **H-4$^V$**, **CH$_2$**N$_3$), 3.15 (dd, $J$ = 11.1, 2.9 Hz, 1H, **H-6$_b$$^{IV}$**), 2.22 (d, $J$ = 8.9 Hz, 1H, 3$^V$-O**H**), 1.61 – 1.47 (m, 4H, **CH$_2$**CH$_2$N$_3$, OCH$_2$**CH$_2$**), 1.35 (m, 5H, 6$^I$-C**H$_3$**, **CH$_2$**CH$_2$CH$_2$N$_3$), 1.17 (d, $J$ = 6.0 Hz, 3H, 6$^{III}$-C**H$_3$**), 0.96 (d, $J$ = 6.1 Hz, 3H, 6$^V$-C**H$_3$**); **$^{13}$C NMR** (126 MHz, CDCl$_3$): δ 138.88 (Ar**C**CH$_2$, OBn), 138.52 (Ar**C**CH$_2$, OBn), 138.45 (Ar**C**CH$_2$, OBn), 138.40 (Ar**C**CH$_2$, OBn), 138.34 (Ar**C**CH$_2$, OBn), 138.17 (Ar**C**CH$_2$, OBn), 138.03 (Ar**C**CH$_2$, OBn), 137.89 (Ar**C**CH$_2$, OBn), 137.87 (Ar**C**CH$_2$, OBn), 137.80 (Ar**C**CH$_2$, OBn), [128.47, 128.34, 128.29, 128.27, 128.22, 128.16, 128.12, 128.09, 128.02, 127.96, 127.85, 127.76, 127.69, 127.63, 127.61, 127.58, 127.55, 127.51, 127.44, 127.36, 127.31, 127.28, 127.17 (OBn)], 98.32 (**C-1$^{III}$**), 98.08 (**C-1$^I$**), 97.37 (**C-1$^V$**), 95.37 (**C-1$^{II}$**), 93.29(**C-1$^{IV}$**), 82.37 (**C-4$^V$**), 80.90 (**C-3$^I$**), 80.30 (**C-4$^I$**), 80.17 (**C-2$^{IV}$**, **C-3$^{IV}$**), 80.02 (**C-2$^{II}$**), 79.56 (**C-4$^{III}$**), 78.39 (**C-2$^V$**), 76.83 (**C-3$^I$**), 75.85 (**C-2$^I$**), 75.52 (Ph**C**H$_2$, 3$^{II}$-OBn), 75.35 – 75.10 (**C-3$^{III}$**, Ph**C**H$_2$, 3$^{IV}$-OBn; Ph**C**H$_2$, 4$^I$-OBn; Ph**C**H$_2$, 4$^{III}$-OBn), 74.98 (**C-4$^{IV}$**), 74.91 (**C-2$^{III}$**), 74.86 (Ph**C**H$_2$, 2$^{III}$-OBn), 74.44 (**C-4$^{II}$**), 73.42 (Ph**C**H$_2$, 6$^{IV}$-OBn), 73.28 (Ph**C**H$_2$, 2$^{II}$-OBn), 73.22 (Ph**C**H$_2$, 6$^{II}$-OBn), 73.18 (Ph**C**H$_2$, 2$^I$-OBn), 73.05 (Ph**C**H$_2$, 2$^{IV}$-OBn), 72.86 (Ph**C**H$_2$, 2$^{III}$-OBn), 72.43 (Ph**C**H$_2$, 2$^V$-OBn), 71.40 (**C-3$^V$**), 70.70 (**C-5$^{II}$**), 70.23 (**C-5$^{IV}$**), 69.14 (**C-5$^{III}$**), 68.93 (**C-6$^{II}$**), 68.67 (**C-6$^{IV}$**), 68.25 (**C-5$^I$**), 67.86 (**C-5$^V$**), 67.24 (O**C**H$_2$CH$_2$), 51.25 (**C**H$_2$N$_3$), 28.98 (OCH$_2$**C**H$_2$), 28.60 (**C**H$_2$CH$_2$N$_3$), 23.36 (**C**H$_2$CH$_2$CH$_2$N$_3$), 18.08 (**C-6$^I$**), 17.99(**C-6$^{III}$**), 17.85 (**C-6$^V$**); **HR-ESI-TOF/MS (m/z):** [M+Na]$^+$ calcd. for C$_{119}$H$_{133}$N$_3$O$_{23}$Na, 1994.92275; found, 1994.92630.

**5-Aminopentyl [α-ʟ-rhamnopyranosyl]-(1→4)-[α-ᴅ-glucopyranosyl]-(1→3)-α-ʟ-rhamnopyranoside HCl salt (28).** Trisaccharide **24** (55 mg, 45 μmol, 1.0 eq) was dissolved in THF:$^t$BuOH:PBS buffer pH4 (4.0 mL, 6:1:3 v/v). The solution was purged with argon. Pd/C (Evonik Noblyst, 0.30 g, 10% wt) was suspended in DMF:H$_2$O (1.0 mL, 4:1 v/v). Concentrated HCl (0.20 mL) was added, after which the suspension was stirred for 15 min. The catalyst was filtered, washed with mQ H$_2$O and added to the trisaccharide solution. The mixture was purged with H$_2$ and subsequently stirred vigorously, under a H$_2$ atmosphere, for 24 hrs. The mixture was filtered over celite. The celite was washed with mQ H$_2$O, after which the filtrate was evaporated *in vacuo*. The residue was purified using size-exclusion chromatography (Biorad Bio-gel P2 support), yielding deprotected trisaccharide **28** as a white solid (22 mg, 39 μmol, 86%).

R$_f$ = 0.0 (H$_2$O:ACN, 20:80 v/v); **$^1$H NMR** (500 MHz, D$_2$O; solvent peak ref'd to 4.79): δ 5.09 (d, $J$ = 3.8 Hz, 1H, **H-1$^{II}$**), 4.90 (d, $J$ = 1.8 Hz, 1H, **H-1$^{III}$**), 4.87 (d, $J$ = 1.9 Hz, 1H, **H-1$^I$**), 4.15 (dd, $J$ = 3.2, 2.0 Hz, 1H, **H-2$^I$**), 4.11 – 4.03 (m, 2H, **H-5$^{II}$**, **H-5$^{III}$**), 4.01 (dd, $J$ = 3.4, 1.8 Hz, 1H, **H-2$^{II}$**), 3.88 (t, $J$ = 9.5 Hz, 1H, **H-3$^{II}$**), 3.84 – 3.80 (m, 2H, **H-3$^I$**, **H-6$_a$$^{II}$**), 3.80 – 3.71 (m, 4H, **H-3$^{III}$**, **H-5$^I$**, **H-6$_b$$^{II}$**, OCH$_a$H$_b$CH$_2$), 3.66 – 3.54 (m, 4H, **H-2$^{II}$**, **H-4$^I$**, **H-4$^{II}$**, OCH$_a$**H$_b$**CH$_2$), 3.48 (t, $J$ = 9.7 Hz, 1H, **H-4$^{III}$**), 3.03 (dd, $J$ = 8.6, 6.7 Hz, 2H, C**H$_2$**N$^+$H$_3$), 1.77 – 1.65 (m, 4H, C**H$_2$**CH$_2$N$^+$H$_3$, OCH$_2$C**H$_2$**), 1.53 – 1.43 (m, 2H, C**H$_2$**CH$_2$CH$_2$N$^+$H$_3$), 1.33 (d, $J$ = 6.4 Hz, 3H, 6$^I$-C**H$_3$**), 1.29 (d, $J$ = 6.3 Hz, 3H, 6$^{III}$-C**H$_3$**); **$^{13}$C NMR** (126 MHz, D$_2$O): δ 100.91 (**C-1$^{III}$**), 99.39 (**C-1$^I$**), 95.57 (**C-1$^{II}$**), 77.25 (**C-4$^{II}$**), 75.95 (**C-3$^I$**), 72.00 (**C-4$^{III}$**), 71.69 (**C-2$^{II}$**), 71.65 (**C-3$^{II}$**), 70.85 (**C-5$^{II}$**), 70.50 (**C-2$^{III}$**), 70.32 (**C-4$^I$**), 70.28 (**C-3$^{III}$**), 69.17 (**C-5$^{III}$**), 68.80 (**C-5$^I$**), 67.59 (-O**C**H$_2$CH$_2$-), 66.99 (**C-2$^I$**), 59.98 (**C-6$^{II}$**), 39.47 (**C**H$_2$N$^+$H$_3$), 28.10 (OCH$_2$**C**H$_2$), 26.60 (**C**H$_2$CH$_2$N$^+$H$_3$), 22.52 (**C**H$_2$CH$_2$CH$_2$N$^+$H$_3$), 16.77 (**C-6$^I$**), 16.56 (**C-6$^{III}$**); **HR-ESI-TOF/MS (m/z):** [M + H]$^+$ calcd. for C$_{23}$H$_{44}$N$_1$O$_{16}$, 558.27618; found, 558.27548.

**5-Aminopentyl [α-ᴅ-glucopyranosyl]-(1→3)-[α-ʟ-rhamnopyranosyl]-(1→4)-[α-ᴅ-glucopyranosyl]-(1→3)-α-ʟ-rhamnopyranoside HCl salt (29).** Tetrasaccharide **25** (88 mg, 49 μmol, 1.0 eq) was dissolved in THF:$^t$BuOH:PBS buffer pH4 (5.0 mL, 6:1:3 v/v). The solution was purged with argon. Pd/C (Evonik Noblyst, 300 mg, 10% wt) was suspended in DMF:H$_2$O (1.0 mL, 4:1 v/v). Concentrated HCl (0.20 mL) was added, after which the suspension was stirred for 15 min. The catalyst was filtered, washed with mQ H$_2$O and added to the tetrasaccharide solution. The mixture was purged with H$_2$ and subsequently stirred vigorously, under a H$_2$ atmosphere, for 114 hrs. The mixture was filtered over celite. The celite was washed with mQ H$_2$O, after which the filtrate was evaporated *in vacuo*. The residue was purified using size-exclusion chromatography (Biorad Bio-gel P2 support), yielding deprotected tetrasaccharide **29** as a white solid (32 mg, 44 μmol, 91%).

R$_f$ = 0.0 (H$_2$O:ACN, 20:80 v/v); **$^1$H NMR** (500 MHz, D$_2$O; solvent peak ref'd to 4.79): δ 5.10 – 5.07 (m, 2H, **H-1$^{II}$**, **H-1$^I$**), 4.95 (d, $J$ = 2.0 Hz, 1H, **H-1$^{III}$**), 4.86 (d, $J$ = 1.9 Hz, 1H, **H-1$^I$**), 4.21 (t, $J$ = 2.6 Hz, 1H, **H-2$^{II}$**), 4.15 (t, $J$ = 2.5 Hz, 1H, **H-2$^I$**), 4.12 – 4.06 (m, 2H, **H-5$^{II}$**, **H-5$^{III}$**), 3.99 (dt, $J$ = 10.2, 3.4 Hz, 1H, **H-5$^{IV}$**), 3.90 – 3.71 (m, 10H, **H-3$^I$**, **H-3$^{II}$**, **H-3$^{III}$**, **H-3$^{IV}$**, **H-5$^I$**, **H-6$_a$$^{II}$**, **H-6$_b$$^{II}$**, **H-6$_a$$^{IV}$**, **H-6$_b$$^{IV}$**, OCH$_a$H$_b$CH$_2$), 3.66 – 3.54 (m, 6H, **H-2$^{II}$**, **H-2$^{IV}$**, **H-4$^I$**, **H-4$^{II}$**, **H-4$^{III}$**, OCH$_a$**H$_b$**CH$_2$), 3.49 (t, $J$ = 9.6 Hz, 1H, **H-4$^{IV}$**), 3.02 (t, $J$ = 7.6 Hz, 2H, C**H$_2$**N$^+$H$_3$), 1.76 – 1.64 (m, 4H, C**H$_2$**CH$_2$N$^+$H$_3$, OCH$_2$C**H$_2$**), 1.53 – 1.42 (m, 2H, C**H$_2$**CH$_2$CH$_2$N$^+$H$_3$), 1.33 (d, $J$ = 6.3 Hz, 3H, 6$^I$-C**H$_3$**), 1.31 (d, $J$ = 6.3 Hz, 3H, 6$^{III}$-C**H$_3$**); **$^{13}$C NMR** (126 MHz, D$_2$O): δ 100.46 (**C-1$^{II}$**), 99.36 (**C-1$^I$**), 95.62 (**C-1$^{IV}$**), 95.56 (**C-1$^{II}$**), 77.22 (**C-4$^{II}$**), 75.94 (**C-3$^I$**), 75.56 (**C-3$^{III}$**), 72.97 (**C-3$^{IV}$**), 71.71 (2 C, **C-2$^{II}$**, **C-5$^{IV}$**), 71.63 (**C-3$^{II}$**), 71.46 (**C-2$^{IV}$**), 70.81 (**C-5$^{II}$**), 70.28 (**C-4$^I$**), 70.21 (**C-4$^{III}$**), 69.37 (**C-4$^{IV}$**), 69.25 (**C-5$^{III}$**), 68.78 (**C-5$^I$**), 67.57 (O**C**H$_2$CH$_2$), 67.17 (**C-2$^{III}$**), 66.98 (**C-2$^I$**), 60.25 (**C-6$^{IV}$**), 59.95 (**C-6$^{II}$**), 39.44 (**C**H$_2$N$^+$H$_3$), 28.08 (OCH$_2$**C**H$_2$), 26.59 (**C**H$_2$CH$_2$N$^+$H$_3$), 22.51 (**C**H$_2$CH$_2$CH$_2$N$^+$H$_3$), 16.74 (**C-6$^I$**), 16.66 (**C-6$^{III}$**); **HR-ESI-TOF/MS (m/z):** [M + H]$^+$ calcd. for C$_{29}$H$_{54}$N$_1$O$_{19}$, 720.32900; found, 720.32780.

**5-Aminopentyl [α-ʟ-rhamnopyranosyl]-(1→4)-[α-ᴅ-glucopyranosyl]-(1→3)-[α-ʟ-rhamnopyranosyl]-(1→4)-[α-ᴅ-glucopyranosyl]-(1→3)-α-ʟ-rhamnopyranoside HCl salt (30).** Pentasaccharide **27** (62 mg, 31 μmol, 1.0 eq) was dissolved in THF:$^t$BuOH:PBS buffer pH4 (6.0 mL, 8:1:3 v/v). The solution was purged with argon. Pd/C (Evonik Noblyst, 200 mg, 10% wt) was suspended in DMF:H$_2$O (1.0 mL, 4:1 v/v). Concentrated HCl (0.20 mL) was added, after which the suspension was stirred for 15 min. The catalyst was filtered, washed with mQ H$_2$O and added to the pentasaccharide solution. The mixture was purged with H$_2$ and subsequently stirred vigorously, under a H$_2$ atmosphere, for 70 hrs. The mixture was filtered over celite. The celite was washed with mQ H$_2$O, after which the filtrate was evaporated *in vacuo*. The residue was purified using size-exclusion chromatography (Biorad Bio-gel P2 support), yielding deprotected pentasaccharide **30** as a white solid (24 mg, 28 μmol, 89%).

R$_f$ = 0.00 (H$_2$O:ACN, 20:80 v/v). **$^1$H NMR** (500 MHz, D$_2$O; solvent peak ref'd to 4.79): δ 5.11 – 5.07 (m, 2H, **H-1$^{II}$**, **H-1$^I$**), 4.95 (d, $J$ = 1.8 Hz, 1H, **H-1$^{III}$**), 4.91 (d, $J$ = 1.7 Hz, 1H, **H-1$^V$**), 4.87 (d, $J$ = 1.9 Hz, 1H, **H-1$^I$**), 4.21 (t, $J$ = 2.5 Hz, 1H, **H-2$^{III}$**), 4.15 (t, $J$ = 2.5 Hz, 1H, **H-2$^I$**), 4.13 – 4.03 (m, 4H, **H-5$^{II}$**, **H-5$^{III}$**, **H-5$^{IV}$**, **H-5$^V$**), 4.01 (t, $J$ = 2.6 Hz, 1H, **H-2$^V$**), 3.92 – 3.85 (m, 2H, **H-3$^{II}$**, **H-3$^{IV}$**), 3.86 – 3.71 (m, 9H, **H-3$^I$**, **H-3$^{III}$**, **H-3$^V$**, **H-5$^I$**, **H-6$_a$$^{II}$**, **H-6$_b$$^{II}$**, **H-6$_a$$^{IV}$**, **H-6$_b$$^{IV}$**, OCH$_a$H$_b$CH$_2$), 3.67 – 3.54 (m, 7H, **H-2$^{II}$**, **H-2$^{IV}$**, **H-4$^I$**, **H-4$^{II}$**, **H-4$^{III}$**, **H-4$^{IV}$**, OCH$_a$**H$_b$**CH$_2$), 3.48 (t, $J$ = 9.7 Hz, 1H, **H-4$^V$**), 3.03 (t, $J$ = 7.6 Hz, 2H, C**H$_2$**N$^+$H$_3$), 1.77 – 1.63 (m, 4H, C**H$_2$**CH$_2$N$^+$H$_3$, OCH$_2$C**H$_2$**), 1.54 – 1.42 (m, 2H, C**H$_2$**CH$_2$CH$_2$N$^+$H$_3$), 1.33 (d, $J$ = 6.3 Hz, 3H, 6$^I$-C**H$_3$**), 1.31 (d, $J$ = 6.4 Hz, 3H, 6$^{III}$-C**H$_3$**), 1.29 (d, $J$ = 6.3 Hz, 3H, 6$^V$-C**H$_3$**); **$^{13}$C NMR** (126 MHz, D$_2$O): δ 100.89 (**C-1$^V$**), 100.48 (**C-1$^{III}$**), 99.37 (**C-1$^I$**), 95.55 (**C-1$^{II}$, C-1$^{IV}$**), 77.25 (**C-4$^{II}$**), 77.14 (**C-4$^{IV}$**), 75.93 (**C-3$^I$**), 75.73 (**C-3$^{III}$**), 71.98 (**C-4$^V$**), 71.69 (2 C, **C-2$^{II}$**, **C-2$^{IV}$**), 71.64 (**C-3$^{II}$**, **C-3$^{IV}$**), 70.82 (2 C, **C-5$^{II}$**, **C-5$^{IV}$**), 70.48 (**C-2$^V$**), 70.31 - 70.26 (2 C, **C-4$^I$**, **C-4$^{III}$**), 70.23 (**C-3$^V$**), 69.27 (**C-5$^{III}$**), 69.15 (**C-5$^V$**), 68.78 (**C-5$^I$**), 67.58 (O**C**H$_2$CH$_2$), 67.24 (C-2$^{III}$), 66.98 (**C-2$^I$**), 59.96 - 59.88 (2C, **C-6$^{II}$**, **C-6$^{IV}$**), 39.45 (**C**H$_2$N$^+$H$_3$), 28.09 (OCH$_2$**C**H$_2$), 26.59 (**C**H$_2$CH$_2$N$^+$H$_3$), 22.51 (**C**H$_2$CH$_2$CH$_2$N$^+$H$_3$), 16.76 (**C-6$^I$**), 16.67 (**C-6$^{III}$**), 16.55 (**C-6$^V$**); **HR-ESI-TOF/MS (m/z):** [M + H]$^+$ calcd. for C$_{35}$H$_{64}$N$_1$O$_{23}$, 866.38691; found, 866.38778.

## Data availability

All instrumental details, formulas, Supplementary Figs. kinetics, VT NMR procedures and studies, (synthetic) experimental procedures, NMR spectra, and supplementary NMR data can be found in the supplementary information. The authors declare that the data supporting the findings of this study are available within the paper and its supplementary method and data files. All other data, such as raw NMR

files, are deposited at the Institute for Molecules and Materials (Radboud University Nijmegen) and are available from the corresponding authors on request.

## Code availability

IRIS data acquirement and the generation of computational IR spectra have been described in the methods section. The authors declare that all formulas and equations used for calculating reaction kinetics can be found in the SI (Supplementary Pages 8-12).

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

## Acknowledgements

This work was supported by a VIDI grant (192.070) awarded to T.J.B. We gratefully acknowledge the Nederlandse Organisatie voor Wetenschappelijk Onderzoek (NWO) for the support of the FELIX Laboratory through the research program "National Roadmap Grootschalige Wetenschappelijke Infastructuur" 184.034.022. This project received funding from NWO Rekentijd for the computational resources (2021.055). We kindly thank Pepijn Geutjes, dr. Tom Bloemberg, and Luuk van Summeren from the Faculty of Science at the Radboud University Nijmegen for making their Bruker 300 MHz NMR spectrometer available to conduct this research on.

## Author contributions

P.H.M., F.B., F.F.J.K., P.B.W. and T.J.B. designed the experiments and were involved in scientific discussions. P.H.M. performed the synthesis and analysis of compounds. F.B. conducted the gas-phase IRIS studies. F.F.J.K. executed the solution-phase NMR studies. B.B., S.J.R.C. and H.R.A. assisted in synthetic experiments. K.J.H., G.B., J.M., J.O. were critical in the gas-phase studies. P.H.M., F.B., F.F.J.K., K.J.H., J.M., J.O., P.B.W. and T.J.B. critically reviewed the paper. P.B.W. and T.J.B. directed the project.

## Competing interests

The authors declare no competing interests.
