## [Peer Review File · Nature Communications]

REVIEWER COMMENTS

Reviewer #1 (Remarks to the Author):

Boltje and co-workers, in this manuscript, reported the characterization of rhamnose 1,3-bridged dioxanium ions derived from 3-O-p-anisoylated rhamnosyl donor. The preferred glycosylation intermediate in the gas-phase has been verified by the authors through the combined use of quantum-chemical calculations and infrared ion spectroscopy (IRIS). They also determined the structure and exchange kinetics of the highly-reactive species in the solution-phase using various testing methods. Then, they applied the C-3 acylated rhamnosyl donor to the synthesis of several bacterial oligosaccharides. Overall, this work validates the involvement effect of the C3 acyl group in the rhamnose system and provides a good example for the investigation of glycosylation mechanism. However, not only the concept but the methodologies adopted in the paper have been previously reported by the authors. Furthermore, the stereodirecting effect of C3-esters on the glycosylation stereochemistry of rhamnosyl thioglycoside donors has already been demonstrated (see ref 6: Eur. J. Org. Chem. 2019, 2019, 6377). Thus, the novelty of this article is insufficient for publication in Nature Comm..

Reviewer #2 (Remarks to the Author):

Arguably the most important reaction in oligosaccharide synthesis is the union of two carbohydrate building blocks in a glycosylation reaction with a controlled anomeric stereoselectivity. For half a century, the most prominent finding was the possibility of preparing stereoselectivity 1,2-trans-glycopyranosides by the use of neighboring group participation (NGP), which has been of fundamental importance to carbohydrate chemistry. However, the potential participating effect of esters located in more remote positions of glycopyranosyl donors has not been unambiguously established, and this has led to a long-term debate. The highly reactive and unstable nature of the bridged dioxanium or dioxepanium intermediates under standard glycosylation conditions severely complicates their characterization. Recently, the authors reported the spectroscopic evidence of bridged intermediates resulting from the NGP of C-3 and C-4 esters on several glycosyl donors using gas-phase infrared ion spectroscopy (IRIS) with density functional theory calculations and a systematic series of glycosylation reactions. In the current manuscript, the authors reported a workflow that integrated their previous work with chemical exchange saturation transfer (CEST), and extensive NMR studies to characterize rhamnosyl 1,3-bridged dioxanium ions derived from C-3 p-anisoyl esterified donors. This combined approach provides a robust basis for elucidating highly reactive intermediates in glycosylation reactions.

I would like to recommend the publication of the manuscript after minor revision.

There are several typos in Figure 6. The target glycans should be 28 made from 24, and 29 from 25, respectively.

It would be helpful if the authors provided more information on why they selected the glycans with the α -(1,3)-rhamnopyranosyl- α -(1,4)-glucopyranosyl repeating unit as their synthetic targets. In other words, the authors should elaborate on how the current findings enabled the synthesis. Based on my experience, there are other methods of making choices, and it would be useful to explore those as well. For example, the rhamnopyranoside with expected α -selectivity could also be obtained from a rhamnopyranosyl donor with the acyl-protecting groups on 2 and 4-OHs and an NAP on 3-OH, due to the presence of the promised 2-neighboring participation group.

Reviewer #3 (Remarks to the Author):

The manuscript by Moons and coworkers describes a thorough study of the reactive intermediates formed during rhamnosylations reactions and exploit their knowledge in the synthesis of a bacterial oligosaccharide with a disaccharide repeating unit.

As mentioned by the authors their method have been used in other related studies but not before i the rhamnosyl system.

The manuscript is overall well-written, the science is sound, of high importance and non-trivial to conduct. The study could be publishable in Nature Commun. after addressing some concerns.

Figure 1 and Table S1 (SI) report on the anomeric selectivity, but it is not clear to this reviewer that the structure of the acceptor is. This should be made very clear to the reader.

The authors correctly compare their results to the glycosylation with the 3-O-benzylated donor (2), but not the 3-O-benzoylated donor. This ought to be included in the study in order to see the significance of the p-methoxy group.

This reviewer also strongly suggest the authors to write the reaction condition on the reaction arrows instead of the "old fashioned way" as scheme legends as this doesn't facility easy reading.

Reviewer #4 (Remarks to the Author):

The manuscript submitted by Boltje reported their comprehensive study on rhamnosyl dioxonium ions. This work is rigorous and interesting and the manuscript is well written. I therefore enjoyed reading this paper, which has well explained and provided good evidences for some very important but debatable issues in carbohydrate chemistry, including long range neighboring group participating effect and the behaviors of the corresponding 1,3-bridged dioxanium ions. I am supportive to the publication of this work after some minor revision.

1. The literature references have room to improve. Since Pagel also uses IR ion spectra to study dioxanium ion and long range neighboring group participation, it is more proper if the following works are cited. Eike Mucha et al. *Nat. Commun.* 2018, 9, 4174.

Eike Mucha et al. *Angew. Chem. Int. Ed.* 2020, 59, 6166-6171.

Márkó Grabarics et al. *Chem. Rev.* 2022, 122, 7840-7908.

Chun-Wei Chang et al. Preprint 2023, DOI: 10.21203/rs.3.rs-3512691/v1. Also for the experiments in Figure 1, it is clear that using NIS/TfOH and Ph₂SO/Tf₂O/BSP as promoters showed different results. According to a recent work "*Sci. Adv.* 2023, 9(42), eadk0531", glycosyl iodide could have been involved in these study. The authors should cite this paper to address this difference.

2. Please double check the reference section. The formats of the references should be unified. Some information of ref 10 must be missing.

3. I am wondering why the authors rotate the structures of the 1,3-bridged dioxanium ions by 180 degrees, for example 1d in figure 1. While all the sugar structures with their C1 in the right and C4 in the left, I can not understand why only the 1,3-dioxanium ions have the structures drew with their C1 in the left and C4 in the right? This not only confuses the readers but also themselves. In page 6, the second paragraph, the authors mentioned "¹H-¹H COSY NMR suggests that the dioxanium ion adopts a 1C₄-chair conformation because...(Figure 4B)"; however, if the structure in Figure 4B was drew in a normal way, it is clearly a 4C₁. The structure of L-rhamnoside itself is normally regarded as 1C₄; therefore after the chair flip owing to the formation of 1,3 bridge, it should be 4C₁.

4. In page 5, the bottom of the second paragraph, the authors mentioned "Moreover, an excellent match...". Since so far the "match" is still based on human interpretation, there is no way to define how much percentage match is "excellent". For example, Figure 3D doesn't look a great match to me. Therefore, I suggest this can be revised to something like "a better match of Figure 3B,D as compared to Figure 3A,C".

5. Minorly, step c is missing in Figure 6.

6. Table 1 seems redundant and out of the scope of this manuscript. Moving it to the SI is recommended.

REMARKS TO REVIEWERS.

Reviewer #1 (Remarks to the Author):

Boltje and co-workers, in this manuscript, reported the characterization of rhamnose 1,3-bridged dioxanium ions derived from 3-O-p-anisoylated rhamnosyl donor. The preferred glycosylation intermediate in the gas-phase has been verified by the authors through the combined use of quantum-chemical calculations and infrared ion spectroscopy (IRIS). They also determined the structure and exchange kinetics of the highly-reactive species in the solution-phase using various testing methods. Then, they applied the C-3 acylated rhamnosyl donor to the synthesis of several bacterial oligosaccharides. Overall, this work validates the involvement effect of the C3 acyl group in the rhamnose system and provides a good example for the investigation of glycosylation mechanism. However, not only the concept but the methodologies adopted in the paper have been previously reported by the authors. Furthermore, the stereodirecting effect of C3-esters on the glycosylation stereochemistry of rhamnosyl thioglycoside donors has already been demonstrated (see ref 6: Eur. J. Org. Chem. 2019, 2019, 6377). Thus, the novelty of this article is insufficient for publication in Nature Comm..

We thank the reviewer for the comments. The field investigating the glycosylation reaction mechanism has mostly produced reasonable hypotheses based on indirect experimental evidence (including the work presented in ref 6). Rhamnosyl dioxanium ions have never been detected or characterized before. In addition, what has been missing in the field in our opinion, are methods to unequivocally prove glycosylation reaction mechanisms based on direct experimental evidence. Therefore, we consider a major novel component of the presented work the development of an integrated workflow to 1. Experimentally characterize fleeting reaction intermediates; 2. Measure the kinetics of reaction intermediate interconversion; 3. Establish the mechanism of reaction intermediate interconversion; and 4. Explore the robustness and reliability of new methodology in the context of oligosaccharide synthesis. Hence, this work outlines a path to unequivocally prove the existence of fleeting glycosylation reaction intermediates by providing direct experimental evidence using this workflow. In our opinion this is a novel and impactful development to finally move the field towards the complete understanding of the glycosylation reaction mechanism based on experimental evidence.

Reviewer #2 (Remarks to the Author):

Arguably the most important reaction in oligosaccharide synthesis is the union of two carbohydrate building blocks in a glycosylation reaction with a controlled anomeric stereoselectivity. For half a century, the most prominent finding was the possibility of preparing stereoselectivity 1,2-trans-glycopyranosides by the use of neighboring group participation (NGP), which has been of fundamental importance to carbohydrate chemistry. However, the potential participating effect of esters located in more remote positions of glycopyranosyl donors has not been unambiguously established, and this has led to a long-term debate. The highly reactive and unstable nature of the bridged dioxanium or dioxepanium intermediates under standard glycosylation conditions severely complicates their characterization. Recently, the authors reported the spectroscopic evidence of bridged intermediates resulting from the NGP of C-3 and C-4 esters on several glycosyl donors using gas-phase infrared ion spectroscopy (IRIS) with density functional theory calculations and a systematic series of glycosylation reactions. In the current manuscript, the authors reported a workflow that integrated their previous work with chemical exchange saturation transfer (CEST), and extensive NMR studies to characterize rhamnosyl 1,3-bridged dioxanium

ions derived from C-3 p-anisoyl esterified donors. This combined approach provides a robust basis for elucidating highly reactive intermediates in glycosylation reactions. I would like to recommend the publication of the manuscript after minor revision.

There are several typos in Figure 6. The target glycans should be 28 made from 24, and 29 from 25, respectively.

We thank the reviewer for pointing this out. We have changed the numbering of the target glycans. They now correspond with the numbering in the SI.

It would be helpful if the authors provided more information on why they selected the glycans with the α -(1,3)-rhamnopyranosyl- α -(1,4)-glucopyranosyl repeating unit as their synthetic targets. In other words, the authors should elaborate on how the current findings enabled the synthesis. Based on my experience, there are other methods of making choices, and it would be useful to explore those as well. For example, the rhamnopyranoside with expected α -selectivity could also be obtained from a rhamnopyranosyl donor with the acyl-protecting groups on 2 and 4-OHs and an NAP on 3-OH, due to the presence of the promised 2-neighboring participation group.

We thank the reviewer for the comment. The main motivation for selecting the α -(1,3)-rhamnopyranosyl- α -(1,4)-glucopyranosyl repeating unit was to investigate the application and reliability of the C-3 benzoyl groups to induce stereocontrol and serve as a temporary protecting group. Furthermore, its synthesis has not been explored before and these oligosaccharides are found on pathogenic bacteria; they may ultimately be used to develop vaccines. Motivations for this are provided in the paragraph introducing this synthetic target.

The reviewer rightly points out that 2-O-acyl participation could also be used to construct α -rhamnosyl bonds. Our work now shows that 3-acyl participation really exists and provides α -rhamnosyl bonds with effectively absolute stereocontrol even in the synthesis of a complex oligosaccharide. This means that the construction of α -rhamnosyl bonds can be carried out even in the absence of a C-2 participation group, which provides additional flexibility in the design of synthetic towards oligosaccharides modified at this site (C-2). We have clarified this in the concluding paragraph of the total synthesis part by the addition of the following statements: "Furthermore, the excellent stereo-directing properties of the C-3 acyl group on rhamnosides in the context of complex oligosaccharide synthesis establishes this method as a reliable alternative for the more established C-2 acyl participation. This abrogates the need for a C-2 participating group hence introducing new possibilities in the design and synthesis of oligosaccharide modified at this position."

Reviewer #3 (Remarks to the Author):

The manuscript by Moons and coworkers describes a through study of the reactive intermediates formed during rhamnosylations reactions and exploit their knowledge in the synthesis of a bacterial oligosaccharide with a disaccharide repeating unit.

As mentioned by the authors their method have been used in other related studies but not before i the rhamnosyl system.

The manuscript is overall well-written, the science is sound, of high importance and non-trivial to conduct. The study could be publishable in Nature Commun. after addressing some concerns.

Figure 1 and Table S1 (SI) report on the anomeric selectivity, but it is not clear to this reviewer that the structure of the acceptor is. This should be made very clear to the reader.

We thank the reviewer for the comment. We have changed the figure so that the structure of the acceptor is shown (bottom-left). In addition, we have added "5-Azidopentanol was used as glycosyl acceptor." to the figure's caption. We have also added the latter to the appropriate table legend in the SI (SI: Table S1).

The authors correctly compare their results to the glycosylation with the 3-O-benzylated donor (2), but not the 3-O-benzoylated donor. This ought to be included in the study in order to see the significance of the p-methoxy group.

We thank the reviewer for the comment. We have executed an additional premix-based glycosylation and a preactivation-based glycosylation of donor 3 (3-O-benzoylated donor) using similar reaction conditions as with the other two donors (1 and 2). We have added the results to figure 1, to the appropriate table and figure in the SI (SI: Table S1 and SI: Figure S3). Similarly to the anisoyl group, the benzoyl group provides full α -selectivity in a preactivation glycosylation strategy. With a premix strategy, it gives a $\sim 21:1$ α/β selectivity, compared to $\sim 37:1$ α/β for the anisoyl group. The latter may be explained by the extra stabilization of the cationic species through resonance.

This reviewer also strongly suggest the authors to write the reaction condition on the reaction arrows instead of the "old fashioned way" as scheme legends as this doesn't facilitate easy reading.

We thank the reviewer for the comment. We have added the reagents above the reaction arrows and removed the conditions from the scheme legend. Figure 6 has been altered to facilitate this.

Reviewer #4 (Remarks to the Author):

The manuscript submitted by Boltje reported their comprehensive study on rhamnosyl dioxonium ions. This work is rigorous and interesting and the manuscript is well written. I therefore enjoyed reading this paper, which has well explained and provided good evidences for some very important but debatable issues in carbohydrate chemistry, including long range neighboring group participating effect and the behaviors of the corresponding 1,3-bridged dioxanium ions. I am supportive to the publication of this work after some minor revision.

1. The literature references have room to improve. Since Pagel also uses IR ion spectra to study dioxanium ion and long range neighboring group participation, it is more proper if the following works are cited. Eike Mucha et al. Nat. Commun. 2018, 9, 4174.

Eike Mucha et al. Angew. Chem. Int. Ed. 2020, 59, 6166-6171.

Márkó Grabarics et al. Chem. Rev. 2022, 122, 7840-7908.

Chun-Wei Chang et al. Preprint 2023, DOI: 10.21203/rs.3.rs-3512691/v1. Also for the experiments in Figure 1, it is clear that using NIS/TfOH and Ph₂SO/Tf₂O/BSP as promoters showed different results. According to a recent work "Sci. Adv. 2023, 9(42), eadk0531", glycosyl iodide could have been involved in these study. The authors should cite this paper to address this difference.

We thank the reviewer for the comments. We have added the first three references. In addition, we have added a few extra references on the topic and changed the accompanying sentence:

"Recently, we reported the spectroscopic evidence of bridged intermediates resulting from the NGP of C-3 and C-4 esters on glucosides, galactosides and mannosides using gas-phase infrared ion spectroscopy (IRIS).^[source]"

To:

“Recently, we and others reported the spectroscopic evidence of bridged intermediates, ^[sources] including those resulting from the NGP of C-3 and C-4 esters on glucosides, galactosides and mannosides, using gas-phase infrared ion spectroscopy (IRIS). ^[source]”

We have also cited the work “*Sci. Adv.* 2023, 9(42), eadk0531” at the start of the results section and changed the sentence accordingly:

“The benzylated rhamnosyl donor likely reacts via glycosyl triflate intermediates of the solvent-separated ion pair (SSIP), thus resulting in poor selectivity.”

To:

“While selectivity may be affected by promoter choice, ^[source] the benzylated rhamnosyl donor likely reacts via glycosyl triflate intermediates of the solvent-separated ion pair (SSIP), thus resulting in poor selectivity independent of the promoter system used.”

We have not cited “Chun-Wei Chang et al. Preprint 2023, DOI: 10.21203/rs.3.rs-3512691/v1” because this paper does not address acyl NGP but instead focuses on the glycosyl cations derived from benzylidene-protected donors that do not carry acyl groups.

2. Please double check the reference section. The formats of the references should be unified. Some information of ref 10 must be missing.

We thank the reviewer for the comment. We have checked the reference section, formatted the references according to Nature reference style and updated the missing information.

3. I am wondering why the authors rotate the structures of the 1,3-bridged dioxanium ions by 180 degrees, for example 1d in figure 1. While all the sugar structures with their C1 in the right and C4 in the left, I can not understand why only the 1,3-dioxanium ions have the structures drawn with their C1 in the left and C4 in the right? This not only confuses the readers but also themselves.

We thank the reviewer for the comment. We have chosen to draw the structure like this for visual reasons. As can be seen in the following examples, the usual form of drawing the bridge rhamnosyl cation (right most structure) provides a structure with overlapping bonds, namely with the dioxanium charge, the C-2 benzyl ether and the C-6 group.

We understand that the rotation of the structure with the C-1 to the left is confusing and in fact, unnecessary. We did this to adopt the viewpoint from the bottom of the ring with the C-3 bond in the middle such that the C-2 and C-6 substituents are not overlapping. This is still a slightly unusual viewpoint so we adapted the figures to the proposed ⁴C₁ conformer in the more traditional viewpoint (from the top of the ring system, right most structure). Despite the overlapping bonds, we agree most readers will be

better able to follow the conversion of the rhamnosyl triflate into the dioxanium ion using this presentation.

In page 6, the second paragraph, the authors mentioned "1H-1H COSY NMR suggests that the dioxanium ion adopts a 1C4-chair conformation because...(Figure 4B)"; however, if the structure in Figure 4B was drawn in a normal way, it is clearly a 4C1. The structure of L-rhamnoside itself is normally regarded as 1C4; therefore after the chair flip owing to the formation of 1,3 bridge, it should be 4C1.

We thank the reviewer for pointing this out. We have corrected the typo to ⁴C₁.

4. In page 5, the bottom of the second paragraph, the authors mentioned "Moreover, an excellent match...". Since so far the "match" is still based on human interpretation, there is no way to define how much percentage match is "excellent". For example, Figure 3D doesn't look a great match to me. Therefore, I suggest this can be revised to something like "a better match of Figure 3B,D as compared to Figure 3A,C".

We thank the reviewer for the comment. We have changed the sentence to:

*"Moreover, a better match is presented for the major dioxanium ion stretches (Figure 3B,D; C⁺-C_{Ar}, O-C⁺-O, and C-H_{OMe}) rather than for the major oxocarbenium stretches (Figure 3A,C) for both the benzoyl and the *p*-anisoyl donors."*

5. Minorly, step c is missing in Figure 6.

We thank the reviewer for pointing this out. We have now placed the reaction reagents above the reaction arrows.

6. Table 1 seems redundant and out of the scope of this manuscript. Moving it to the SI is recommended.

We thank the reviewer for the comment. We have moved the table from the article to the SI.

REVIEWERS' COMMENTS

Reviewer #2 (Remarks to the Author):

All the questions I raised have been addressed satisfactorily. The current version of the manuscript is well-balanced and provides a clear explanation and introduction that can be easily understood by readers. Therefore, I recommend that the manuscript should be published in its current format.

Reviewer #3 (Remarks to the Author):

The manuscript and its results have been significantly improved since its last submission and concerns of the reviewers addressed in a satisfactory fashion. This reviewer can recommend acceptance by Nat. Commun.

Reviewer #4 (Remarks to the Author):

The authors have addressed the questions I raised in their previous submission. Therefore, I suggest this manuscript to be accepted for publication.